# Learning the Arrow of Time for Problems in Reinforcement Learning

**Nasim Rahaman**[1,2,3]  **Steffen Wolf**[1]  **Anirudh Goyal**[3]  **Roman Remme**[1]  **Yoshua Bengio**[3,4,5]

[1]Image Analysis and Learning Lab, Ruprecht-Karls-Universität, Heidelberg
[2]Max-Planck Institute for Intelligent Systems, Tübingen
[3]Mila, Montréal

[4]CIFAR Senior Fellow
[5]Canada CIFAR AI Chair

## Abstract

We humans have an innate understanding of the asymmetric progression of time, which we use to efficiently and safely perceive and manipulate our environment. Drawing inspiration from that, we approach the problem of learning an *arrow of time* in a Markov (Decision) Process. We illustrate how a learned arrow of time can capture salient information about the environment, which in turn can be used to measure reachability, detect side-effects and to obtain an intrinsic reward signal. Finally, we propose a simple yet effective algorithm to parameterize the problem at hand and learn an arrow of time with a function approximator (here, a deep neural network). Our empirical results span a selection of discrete and continuous environments, and demonstrate for a class of stochastic processes that the learned arrow of time agrees reasonably well with a well known notion of an arrow of time due to Jordan, Kinderlehrer, and Otto (1998).

## 1 Introduction

The asymmetric progression of time has a profound effect on how we, as agents, perceive, process and manipulate our environment. Given a sequence of observations of our familiar surroundings (e.g. as video frames), we possess the innate ability to predict whether the said observations are ordered *correctly*. We use this ability not just to perceive, but also to act: for instance, we know to be cautious about dropping a vase, guided by the intuition that the act of breaking a vase cannot be undone. This profound intuition reflects some fundamental properties of the world in which we dwell, and in this work we ask whether and how these properties can be exploited to learn a representation that functionally mimics our understanding of the asymmetric nature of time.

The term *Arrow of Time* was coined by the British astronomer Eddington (1929) to denote this inherent asymmetry, which he attributed to the non-decreasing nature of the total thermodynamic entropy of an isolated system, as required by the second law of thermodynamics. Since then, the notion of an arrow of time has been formalized and explored in various contexts, spanning not only physics, but also algorithmic information theory (Zurek, 1989), causal inference (Janzing et al., 2016) and time-series analysis (Janzing, 2010; Bauer et al., 2016).

Broadly, an arrow of time can be thought of as a function that monotonously increases as a system evolves in time. Expectedly, the notion of irreversibility plays a central role in the discourse. In statistical physics, it is posited that the arrow of time (i.e. entropy production) is *driven* by irreversible processes (Prigogine, 1978; Seifert, 2012). To understand how a notion of an arrow of time can be useful in the reinforcement learning context, consider the example of a cleaning robot tasked with moving a box across a room (Amodei et al., 2016). The optimal way of successfully completing the task might involve the robot doing something disruptive, like knocking a vase over (Fig 1). Now on the one hand, such disruptions – or *side-effects* – might be difficult to recover from. In the extreme case, they might be virtually irreversible – say when the vase is broken. On the other hand, irreversibility implies that states with a larger number of broken vases tend to occur in the future, and one should therefore expect an arrow of time (as a scalar function of the state) to assign larger values

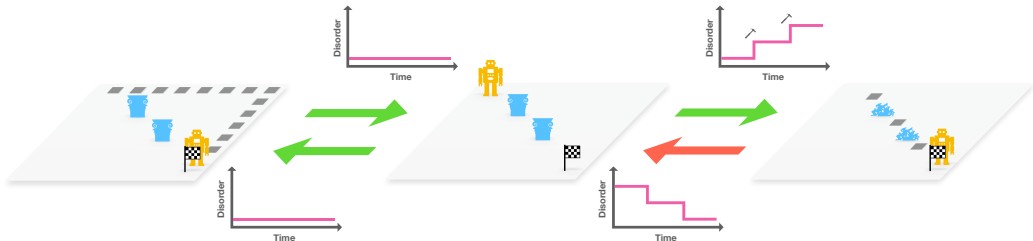

Figure 1: The agent (in orange) is tasked with reaching its goal, the checkered flag (middle frame). It may take the shorter path (right frame), which entails breaking the vases in its way, or it may prefer the safer path (left frame) which is longer but keeps the vases intact. The former path is irreversible, and the initial state is unreachable from the final state (red arrow). On the contrary, the latter path is completely reversible, and the initial state remains reachable from the final state. Now, an arrow of time (pink) measures the disorder, which might help a safe agent decide which path to take.

to states with larger number of broken vases. An arrow of time should therefore quantify the amount of *disorder* in the environment, analogous to the entropy for isolated thermodynamical systems.

Now, one possible application could be to detect and preempt such side-effects, for instance by penalizing policies that significantly increment the arrow of time by executing difficult-to-reverse transitions. But the utility of an arrow of time is more general: it serves as a *directed measure* of reachability. This can be seen by observing that it is more difficult to obtain *order* from *disorder*: it is, after all, difficult to reach a state with a vase intact from one with it broken, rather than vice versa. In this sense, we may say that a state is *relatively unreachable* from another state if an arrow of time assigns a lower value to the former. Further, a directed measure of reachability afforded by an arrow of time can be utilized for deriving an intrinsic reward signal to enable agents to learn complex skills in the absence of external rewards. To see how, consider that an agent tasked with *reversing* the arrow of time (by creating *order* from *disorder*) must in general learn complex skills to achieve its goal. Indeed, gluing together a broken vase will require the agent to learn an array of complex planning and motor skills, which is the ultimate goal of such intrinsic rewards.

In summary, our contributions are the following. **(a)** We propose a simple objective to learn an arrow of time for a Markov (Decision) Process in a self-supervised manner, i.e. entirely from sampled environment trajectories and without external rewards. We call the resulting function (acting on the state) the $h$-potential, and demonstrate its utility and caveats for a selection of discrete and continuous environments. Moreover, we compare the learned $h$-potential to the free-energy functional of stochastic processes – the latter being a well-known notion of an arrow of time (Jordan et al., 1998). While there exist prior work on *detecting* the arrow of time in videos (Pickup et al., 2014; Wei et al., 2018) and time-series data (Peters et al., 2009; Bauer et al., 2016), we believe our work to be the first towards *measuring* it in the context of reinforcement learning. **(b)** We critically and transparently discuss the conceptually rich subtleties that arise before an arrow of time can be practically useful in the RL context. **(c)** We expose how the notions of reachability, safety and curiosity can be unified under the common framework afforded by a learned arrow of time.

## 2 THE $h$-POTENTIAL

Motivated by the preceding discussion, our goal is to learn a function that quantifies the amount of *disorder* in a given environment state, where we say that irreversible state transitions increase *disorder*. In this sense, we seek a function (of the state) that is constant in expectation along *fully reversible* state transitions, but increase in expectation along state transitions that are *less reversible*. To that end, we begin by formally introducing this function, which we call the $h$-potential, as the solution to a functional optimization problem. Subsequently, we critically discuss a few conceptual roadblocks that must be cleared before such a function can be useful in the RL setting.

### 2.1 FORMALISM

Consider a Markov Decision Process (a MDP, i.e. *environment*), and let $\mathcal{S}$ and $\mathcal{A}$ be its state and action spaces respectively. A policy $\pi$ is a mapping from the state space to the space of distributions over actions. Given a state $s \in \mathcal{S}$ sampled from some initial state distribution $p_0$, we may sample

an action $a \in \mathcal{A}$ from the policy $\pi(a|s)$, which in turn can be used to sample another state $s' \in \mathcal{S}$ from the environment dynamics $p(s'|a, s)$. Iterating $N$ more times for a fixed $\pi$, one obtains a sequence of states $(s_0, ..., s_t, ..., s_N)$, which is a realization of the Markov chain (a *trajectory*) with transition probabilities $p_\pi(s_{t+1}|s_t) = \sum_{a \in \mathcal{A}} p(s_{t+1}|s_t, a)\pi(a|s_t)$. We may now define a function $h_\pi : \mathcal{S} \to \mathbb{R}$ as the solution to the following functional objective:

$$\mathcal{J}_\pi[\hat{h}] = \mathbb{E}_{t \sim U(\{0,...,N-1\})}\mathbb{E}_{s_t}\mathbb{E}_{s_{t+1}|s_t}[\hat{h}(s_{t+1}) - \hat{h}(s_t)|s_t] + \lambda \mathcal{T}[\hat{h}]; \quad h_\pi = \arg\max_{\hat{h}} \mathcal{J}_\pi[\hat{h}] \quad (1)$$

where $U(A)$ is the uniform distribution over any set $A$, $\mathbb{E}_t\mathbb{E}_{s_t}\mathbb{E}_{s_{t+1}|s_t}$ is the expectation over all state transitions, $\lambda$ is a scalar coefficient and $\mathcal{T}[\hat{h}]$ is a regularizing term that prevents $\hat{h}$ from diverging within a finite domain. In words: the first term on the right hand side of the first equation above encourages $h_\pi$ to increase in expectation along the sampled trajectories, whereas the second term controls this increase; the two terms are balanced with a coefficient $\lambda$. Informally: if a state transition $s \to s'$ is *fully reversible*, the probability of sampling it equals that of sampling the corresponding reverse transition, $s' \to s$. For such transitions, the pressure on $h_\pi$ to increase along the forward transition ($s \to s'$) is compensated by the counter-pressure for it to increase along the reverse transition ($s' \to s$), or equivalently, decrease along the forward transition. Along such transitions, we should therefore expect $h_\pi$ to remain constant (in expectation). Accordingly, if the forward transition were to be more likely (i.e. if the transition is not fully reversible), we should expect $h_\pi$ to increase (in expectation) in order to satisfy its objective.

The regularizer $\mathcal{T}$ must be chosen to suit the problem at hand, and different choices result in solutions that have different characteristics[1]. Possible choices for $\mathcal{T}$ include (any combination of) the negative of $L_2$ norm $-\|\hat{h}\|_2$, and/or the following trajectory regularizer:

$$\mathcal{T}[\hat{h}] = -\mathbb{E}_{t \sim U(\{0,...,N-1\})}\mathbb{E}_{s_t}\mathbb{E}_{s_{t+1}|s_t}[|\hat{h}(s_{t+1}) - \hat{h}(s_t)|^2|s_t] \quad (2)$$

Intuitively: while the solution $h_\pi$ is required to increase in expectation along trajectories, the trajectory regularizer acts as an contrastive term by penalizing $h_\pi$ for changing at all.

With some effort, the problem defined in Eqn 1 can be approached analytically for toy Markov chains (interested readers may refer to App A for a technical discussion). However, such analytical treatment becomes infeasible for more complex and larger-scale environments with unknown transition probabilities. To tackle such environments, we will cast the functional optimization problem in Eqn 1 to an optimization problem over the parameters of a deep neural network and solve it for a variety of discrete and continuous environments.

## 2.2 SUBTLETIES

In this section, we discuss two conceptually rich subtleties that determine the conditions under which the learned arrow of time ($h$-potential) can be useful in practice.

**The Role of a Policy.** The first subtlety is rooted in the observation that the trajectories $(s_0, ..., s_N)$ are collected by a given but arbitrary policy. However, there may exist policies for which the resulting arrow of time is unnatural, perhaps even misleading. Consider for instance the actions of a practitioner of Kintsugi, the ancient Japanese art of repairing broken pottery. The corresponding policy[2] might cause the environment to transition from a state where the vase is broken to one where it is not. If we learn the $h$-potential on such trajectories, it might be the case that counter to our intuition, states with a larger number of broken vases are assigned smaller values (and the vice versa). Now, one may choose to resolve this conundrum by defining:

$$\mathcal{J}[h] = \mathbb{E}_{\pi \sim U(\Pi)}\mathcal{J}_\pi[h] \quad (3)$$

where $\Pi$ is the set of all policies defined on $\mathcal{S}$, and $U(\Pi)$ denotes a uniform distribution over $\Pi$. The resulting function $h^* = \arg\max\{\mathcal{J}[h] + \lambda\mathcal{T}[h]\}$ would characterize the arrow of time with respect to all possible policies, and one would expect that for a vast majority of such policies, the transition from broken vase to a intact vase is rather unlikely and/or requires highly specialized policies.

---

[1]This is not unlike the case for linear regression: for instance, using Lasso instead of ridge-regression will generally yield solutions that have different properties.

[2]This is analogous to Maxwell's demon in classical thermodynamics.

Unfortunately, determining $h^*$ is not feasible for most interesting applications, given the outer expectation over *all* possible policies. As a compromise, we use (uniformly) random actions to gather trajectories. The simplicity of the corresponding random policy justifies its adoption, since one would expect a policy resembling (say) a Kintsugi artist to be rather complex and not implementable with random actions. In this sense, we ensure that the learned arrow of time characterizes the underlying dynamics of the environment, and not the peculiarities of a particular agent[3]. The price we pay is the lack of adequate exploration in complex enough environments, although this problem plagues most model-based reinforcement learning approaches[4] (cf. Ha & Schmidhuber (2018)). In the following, we assume $\pi$ to be uniformly random and use $h_\pi$ interchangeably with $h$.

**Dissipative Environments.** The second subtlety concerns what we require of environments in which the arrow of time is informative. To illustrate the matter, consider the class of systems[5], a typical instance of which could be a billiard ball moving on a frictionless arena and bouncing (elastically) off the edges (Bunimovich, 2007). The state space comprises the ball's velocity and its position constrained to a *billiard table* (without holes!), where the ball is initialized at a random position on the table. For such a system, it can be seen by time-reversal symmetry that when averaged over a large number of trajectories, the state transition $s \rightarrow s'$ is just as likely as the reverse transition $s' \rightarrow s$. In this case, recall that the arrow of time is expected to remain constant. A similar argument can be made for systems that identically follow closed trajectories in their respective state space (e.g. a frictionless and undriven pendulum). It follows that the $h$-potential must remain constant along the trajectory and that the arrow of time is uninformative. However, for so-called *dissipative* systems, the notion of an arrow of time is pronounced and well studied (Willems, 1972; Prigogine, 1978). In MDPs, dissipative behaviour may arise in situations where certain transitions are irreversible by design (e.g. bricks disappearing in Atari Breakout), or due to partial observability, e.g. for a damped pendulum, the state space does not track the microscopic processes that give rise to friction[6]. Therefore, a central premise underlying the practical utility of learning the arrow of time is that the considered MDP is indeed dissipative, which we shall assume in the following; in Sec 5 (Fig 5b), we will empirically investigate the case where this assumption is violated.

## 3 Applications with Related Work

In this section, we discuss a few applications of the arrow of time, and illustrate how the $h$-potential provides a common framework to unify the notions of reachability, safety and curiosity.

### 3.1 Measuring Reachability

Given two states $s$ and $s'$ in $\mathcal{S}$, the reachability of $s'$ from $s$ measures how difficult it is for an agent at state $s$ to reach state $s'$. The prospect of learning reachability from state-transition trajectories has been explored: in Savinov et al. (2018), the approach taken involves learning a logistic regressor network $g^\theta : \mathcal{S} \times \mathcal{S} \rightarrow [0, 1]$ to predict the probability of states $s'$ and $s$ being reachable to one another within a certain number of steps (of a random policy), in which case $g(s, s') \approx 1$. However, the model $g$ is not *directed*: it does not learn whether $s'$ is more likely to follow $s$, or the vice versa. Instead, our proposal is to derive a directed measure of reachability from $h$-potential by defining a function $\eta : \mathcal{S} \times \mathcal{S} \rightarrow \mathbb{R}$ such that $\eta(s, s') \equiv \eta(s \rightarrow s') := h(s') - h(s)$, where $\eta(s \rightarrow s')$ measures the reachability of state $s'$ from state $s$. This inductive bias on $\eta$ (in form of a functional constraint) induces the following useful properties.

**First**, consider the case where the transition between states $s$ and $s'$ is fully reversible, i.e. when state $s$ is exactly as reachable from state $s'$ as is $s'$ from $s$; we denote such transitions with $s \leftrightarrow s'$. Now, in expectation, we obtain that $h(s') = h(s)$ and consequently, $\eta(s \rightarrow s') = \eta(s' \rightarrow s) = 0$. But if instead the state $s'$ is more likely to follow state $s$ than the vice versa (in expectation over trajectories), we say $s'$ is *more reachable* from $s$ than the vice versa. It follows in expectation that

---

[3]What we do is similar (in spirit) to inverse reinforcement learning the reward function maximized by a random policy (instead of an expert policy), cf. Ng & Russell (2000).

[4]While this is a fundamental problem (App C.3), powerful methods for off-policy learning exist (see Munos et al. (2016) and references therein); however, a full analysis is beyond the scope of the current work.

[5]Precisely: Hamiltonian systems where Liouville's theorem holds and the Hamiltonian is time-independent.

[6]In particular, observe that a dissipative system may or may not be ergodic.

$h(s') > h(s)$, and consequently, $\eta(s \to s') > 0$. Now the inductive bias on $\eta$ as a difference of $h$-potentials automatically implies $\eta(s' \to s) = -\eta(s \to s') < 0$.

**Second**, observe that the reachability measure implemented by $\eta$ is additive by construction: given a trajectory $s_0 \to s_1 \to s_2$, we have that $\eta(s_0 \to s_2) = \eta(s_0 \to s_1) + \eta(s_1 \to s_2)$. As a special case, if we have that $s_0 \leftrightarrow s_1$ and $s_1 \leftrightarrow s_2$ – i.e. if $\eta(s_0 \to s_1) = \eta(s_1 \to s_2) = 0$ – it identically follows that $s_0 \leftrightarrow s_2$, i.e. $\eta(s_0 \to s_2) = 0$. In this case, the inductive bias enables $\eta$ to generalize to the transition $s_0 \leftrightarrow s_2$ even if it is never explicitly sampled by the policy.

**Third**, $\eta$ allows for a *soft* measure of reachability. It measures not only *whether* a state $s'$ is reachable from another state $s$, but also quantifies *how* reachable the former is from the latter. As an example, consider a trajectory $s_0 \to s_1 \to ... \to s_{100}$, where the agent breaks one vase at every state transition. If the $h$-potential increases in constant increments for every vase broken (which we confirm it does in Sec 5), we obtain due to the inductive bias that $\eta(s_0 \to s_{100}) = 100 \cdot \eta(s_0 \to s_1)$. This behaviour is sought-after in the context of AI-Safety (Krakovna et al., 2018; Leike et al., 2017).

Nonetheless, one should be careful when interpreting $\eta$. While the above implies that $\eta(s' \to s) = \eta(s \to s')$ if the transition between states $s$ and $s'$ is fully reversible, the converse can only be guaranteed if the Markov process admits a trajectory between $s$ and $s'$ in *either* direction, i.e. if there exists a trajectory that visits both $s$ and $s'$ (in any order). Observe that this condition much weaker than ergodicity, which requires that the Markov process admit a trajectory from any given state $s$ to *all* other states $s'$. In fact, the discrete environments we investigate in Sec 5 are non-ergodic.

## 3.2 Detecting and Penalizing Side Effects for Safe Exploration

The problem of detecting and avoiding side-effects is well known and crucially important for safe exploration (Moldovan & Abbeel, 2012; Eysenbach et al., 2017; Krakovna et al., 2018; Armstrong & Levinstein, 2017). Broadly, the problem involves detecting and avoiding state transitions that permanently and irreversibly damage the agent or the environment (Leike et al., 2017). As such, it is fundamentally related to reachability, as in the agent is prohibited from taking actions that drastically reduce the reachability between the resulting state and some predefined *safe* state. In Eysenbach et al. (2017), the authors learn a reset policy responsible for resetting the environment to some initial state after the agent has completed its trajectory. The resulting value function of the reset policy indicates when the actual (*forward*) policy executes an irreversible state transition, but at the cost of the added complexity of training a reset policy. In contrast, Krakovna et al. (2018) propose to attack the problem by measuring reachability relative to a safe *baseline policy* – namely by evaluating the reduction in reachability of all environment states from the current state with respect to that from a *baseline state*, where the latter is defined as the state that system would have (counterfactually) been in had the agent acted according to the corresponding *baseline policy*. However, determining the counterfactual baseline state requires a causal model of the environment, which cannot always assumed to be known.

We propose to directly use the reachability measure $\eta$ defined in Section 3.1 to derive a reward term for safe-exploration. Let $r_t$ be some external reward at time-step $t$. The augmented reward is given by:

$$\hat{r}_t = r_t - \beta \cdot \max\{\eta(s_{t-1} \to s_t), 0\} \tag{4}$$

where $\beta$ is a scaling coefficient. In practice, one may replace $\eta$ with $\sigma(\eta)$, where $\sigma$ is a monotonically increasing transfer function (e.g. a step function). Intuitively, transitions $s \to s'$ that are *less reversible* cause the $h$-potential to increase, and the resulting reachability measure $\eta(s \to s')$ is positive in expectation. This incurs a penalty (due to the negative sign), which is reflected in the value function of the agent. Conversely, transitions that are reversible should have the property that $\eta(s \to s') = 0$ (also in expectation), thereby incurring no penalty.

## 3.3 Rewarding Curious Behaviour

In most reinforcement learning applications, the reward function is assumed to be given; however, shaping a good reward function can often prove to be a challenging endeavour. It is in this context that the notion of *curiosity* comes to play an important role (Schmidhuber, 2010; Chentanez et al., 2005; Pathak et al., 2017; Burda et al., 2018; Savinov et al., 2018). One typical approach towards encouraging curious behaviour is to seek *novel* states that surprise the agent (Schmidhuber, 2010;

Pathak et al., 2017; Burda et al., 2018) and use the error in the agent's prediction of future states is used as a curiosity reward. This approach is however known to be susceptible to the so-called noisy-TV problem, wherein an uninteresting source of entropy like a noisy-TV can induce a large curiosity bonus because the agent cannot predict its future state. Savinov et al. (2018) propose to circumvent the noisy-TV problem by defining novelty in terms of (undirected) reachability, wherein states that are easily reachable from the current state are considered less novel.

The $h$-potential and the corresponding reachability measure $\eta$ affords another way of defining a curiosity reward. Say an agent's policy samples a trajectory from state $s$ to $s'$. Now, recall that $\eta(s \rightarrow s')$ takes a positive value if state $s'$ is reachable from $s$ (with respect to a simple reference policy); we therefore encourage the agent policy to sample trajectories where the $\eta(s \rightarrow s')$ is negative, i.e. where $s'$ is less reachable from $s$. In doing so, we encourage the agent to seek states that are otherwise difficult to reach just by chance, and possibly learn useful skills in the process. In other words, we reward the agent for *reversing* the arrow of time (recall that $\eta(s \rightarrow s') < 0$ implies $h(s') < h(s)$). The general form of the corresponding reward is given by:

$$\hat{r}_t = -\eta(s_{t-1} \rightarrow s_t) \tag{5}$$

While the above is independent of the external reward function defined by the environment, the latter might often align with the former: in many environments, the task at hand is to reach the least reachable state. This is readily recognized in classical control tasks like Pendulum, Cartpole and Mountain-Car, where the goal state is often the least reachable. However, if the environment's specified task requires the agent to inadvertently execute irreversible trajectories, it is possible that our proposed reward is less applicable. Furthermore, while the proposed curiosity reward encourages the agent to reach for difficult-to-reach states, it need not provide an incentive to seek out diverse states. In other words: an agent optimizing the proposed reward may seek out the most difficult-to-reach states, but ignore other *interesting* but less difficult-to-reach states in the process (cf. App C.3).

To summarize, we used the $h$-potential to define a directed measure of reachability (Sec 3.1), which then naturally lead to two applications. In the first (Sec 3.2), we obtained a safety penalty by essentially discouraging the agent from *increasing* the $h$-potential by executing difficult-to-reverse transitions. In the second (Sec 3.3), we argued that encouraging the agent to *decrease* the $h$-potential can provide an useful curiosity (intrinsic) reward signal in the absence of external rewards. In this sense, we have illustrated how the framework of a learned arrow of time (i.e. the $h$-potential) unifies the notions of reachability, safety, and curiosity.

# 4 ALGORITHM

In Sec 2, we proposed a general functional objective, and defined the $h$-potential as the solution to the corresponding functional optimization problem. While the problem could be approached analytically with some effort for certain toy Markov chains (see App A), complex environments with unspecified dynamics require a fundamentally different approach. We therefore convert the functional optimization problem in Eqn 1 (right) to one over the parameters $\theta$ of a deep neural network $\hat{h}_\theta$ to obtain the following surrogate problem:

$$\theta^* = \arg\max_\theta \left\{ \mathbb{E}_{t \sim U(\{0,...,N-1\})} \mathbb{E}_{s_t} \mathbb{E}_{s_{t+1}|s_t} [\hat{h}_\theta(s_{t+1}) - \hat{h}_\theta(s_t)|s_t] + \lambda \mathcal{T}[\hat{h}_\theta] \right\} \tag{6}$$

where $\pi$ is a reference policy, i.e. uniform random, and we denote the solution $\hat{h}_{\theta*}$ by $h$. To train the network, the expectations are replaced by their sample estimates. As for the regularizer, recall that its purpose was to prevent $h$ from diverging within a finite domain – this can be achieved by a loss term $\mathcal{T}$ (like the trajectory regularizer in Eqn 2), or by a training constraint like early stopping.

The training algorithm is rather straightforward and can be summarized as follows (please refer to App B for the full algorithm). We first use an offline reference policy (uniform random, in our experiments) to sample trajectories from the environment. Next, we sample a batch of uniformly random state transitions and evaluate the objective in Eqn 6 (by replacing expectations by their sample estimates). We regularize the either by adding the trajectory regularizer to the objective or by using early stopping to terminate the training after a fixed number of iterations. Finally, we optimize the parameters $\theta$ of $\hat{h}_\theta$ to maximize the objective at hand.

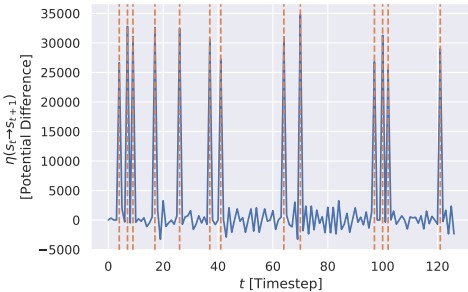

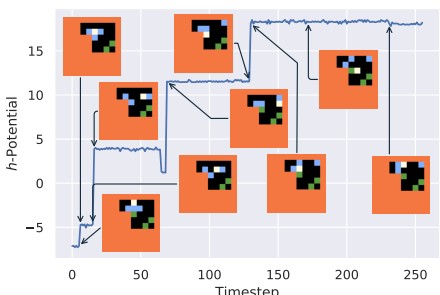

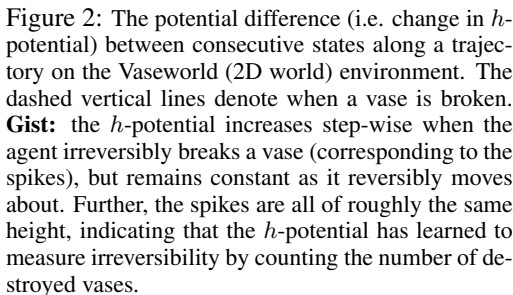

Figure 2: The potential difference (i.e. change in $h$-potential) between consecutive states along a trajectory on the Vaseworld (2D world) environment. The dashed vertical lines denote when a vase is broken. **Gist:** the $h$-potential increases step-wise when the agent irreversibly breaks a vase (corresponding to the spikes), but remains constant as it reversibly moves about. Further, the spikes are all of roughly the same height, indicating that the $h$-potential has learned to measure irreversibility by counting the number of destroyed vases.

Figure 3: The $h$-potential along a trajectory from a random policy, annotated with the corresponding state images on the Sokoban (2D world) environment. The white sprite corresponds to the agent, orange to a wall, blue to a box and green to a goal. **Gist:** the $h$-potential increases sharply as the agent pushes a box against the wall. While it may decrease for a given trajectory (in this case because the agent manages to move a box away from the wall), it increases in expectation over all trajectories (cf. Fig 14 in Appendix C.1.3).

## 5 EXPERIMENTS

In this section, we empirically investigate the $h$-potential that we obtain with the training procedure described in the previous section. First, we show in a 2D-world environment that the $h$-potential learns to measure reachability. Second, we show that the $h$-potential can be used to detect side-effects in the challenging game of Sokoban (Leike et al., 2017). Third, we show on the game of Mountain Car with Friction that the $h$-potential can learn to capture sailent features of the environment, which can be used to formulate an intrinsic reward. We also demonstrate how the $h$-potential fails if the environment is not dissipative, i.e. if the friction is turned off. Finally, we show for a particle undergoing Brownian motion under a potential that in expectation over states, the $h$-potential agrees reasonably well with the Free Energy functional, wherein the latter is known to be an arrow of time (Jordan et al., 1998). Moreover in App C, we show results on three additional environments.

**Measuring Irreversibility.** The environment considered is a $7 \times 7$ 2D world, where cells can be occupied by the agent, the goal and/or a vase (their respective positions are randomly sampled in each episode). If the agent enters a cell with a vase in it, the vase disappears without compromising the agent. In Fig 2, we plot the change in $h$-potential (recall that $\eta(s_t \to s_{t+1}) = h(s_{t+1}) - h(s_t)$) to find that the breaking of a vase (irreversible) corresponds to the $h$-potential increasing in steps of roughly constant size (observe that the spikes attain similar heights), whereas the agent moving around (reversible) does not result in it increasing. This indicates that the $h$-potential has learned to quantify irreversibility instead of merely detecting it by counting the number of broken vases. In App C.1.1, (a) we further investigate the effect of adding temporally-correlated and TV (uncorrelated) noise to the state and find that the $h$-potential is fairly robust to the latter but might get distracted by the former and (b) verify that an agent trained with the safety penalty in Eqn 4 breaks fewer vases (than without).

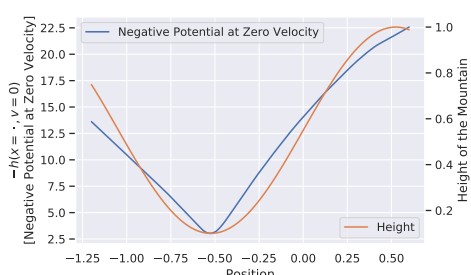

Figure 4: The $h$-potential (for Mountain Car) at zero-velocity plotted against position. Also plotted (orange) is the height profile of the mountain. **Gist:** the $h$-potential approximately recovers the height-profile of the mountain with just trajectories from a random policy.

**Detecting Side-Effects.** Sokoban ("warehouse-keeper") is a challenging puzzle video game, where an agent must push a number of boxes to set goal locations placed on a map. The agent may only push boxes (and not pull), rendering certain moves irreversible - for instance, when a box is pushed

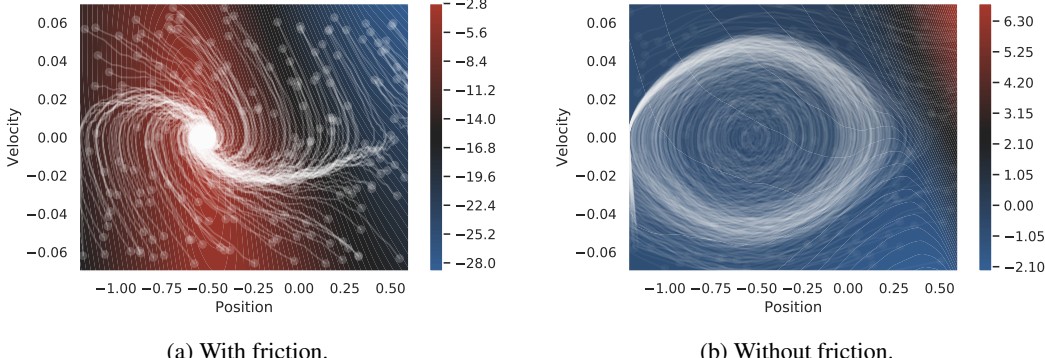

(a) With friction.

(b) Without friction.

Figure 5: The $h$-potential as a function of state (position and velocity) for (continuous) Mountain-Car with and without friction. The overlay shows random trajectories (emanating from the dots). **Gist:** with friction, we find that the state with largest $h$ is one where the car is stationary at the bottom of the valley. Without friction, there is no dissipation and the car oscillates up and down the valley. Consequently, we observe that the $h$-potential is constant (up-to edge effects) and thereby uninformative.

against a wall. Moreover, the task of even determining whether a move is irreversible might be non-trivial, making the problem a good test-bed for detecting side-effects (Leike et al., 2017). In Fig 3, we see that the $h$-potential increases if a box is pushed against a wall (irreversible side-effect) but remains constant if the agent moves about (reversible, even when the agent pushes a box around), demonstrating that the $h$-potential has indeed learned to detect side-effects. For experimental details and additional plots, please refer to App C.1.3.

**Obtaining Intrinsic Reward and the Importance of Dissipativity.** The environment considered shares its dynamics with the well known (continuous) Mountain-Car environment (Sutton & Barto, 2011), but with a crucial amendment: the car is subject to friction. Friction is required to make the environment dissipative and thereby induce an arrow of time (cf. Sec 2.2). Moreover, we initialize the system in a uniform-randomly sampled state to avoid exploration issues (cf. App C.3). In Fig 4, we see that the learned $h$-potential roughly recovers the terrain from random trajectories (i.e. without external rewards), which can now be used to obtain an intrinsic reward signal. Further, Fig 5b illustrates the importance of dissipation (in this case, induced via friction). Details in App C.2.2.

**Comparison with the Free-Energy Functional.** The setting considered is that of a particle (a *random-walker*) undergoing Brownian motion under the influence of a potential field $\Psi(\mathbf{x})$ (where $\mathbf{x}$ denotes the spatial position). We denote the probability of finding the particle at position $\mathbf{x}$ at time $t$ by $\rho(\mathbf{x}, t)$. Now, the dynamics of the corresponding time-dependent random variable (i.e. stochastic process) $\mathbf{X}(t)$ is governed by the stochastic differential equation:

$$dX(t) = -\nabla\Psi(\mathbf{X}(t))dt + \sqrt{2\beta^{-1}}d\mathbf{W}(t) \quad (7)$$

where $\mathbf{W}(t)$ is the standard Wiener process (i.e. $d\mathbf{W}(t)$ is white-noise) and $\beta^{-1}$ is a temperature parameter. The Free-Energy functional $F$ is now defined as:

$$F[\rho(\cdot, t)] = \mathbb{E}_{\mathbf{x}\sim\rho(\cdot,t)}\left[\Psi(\mathbf{x})\right]$$
$$+\beta^{-1}\mathbb{E}_{\mathbf{x}\sim\rho(\cdot,t)}\left[\log\rho(\mathbf{x}, t)\right] \quad (8)$$

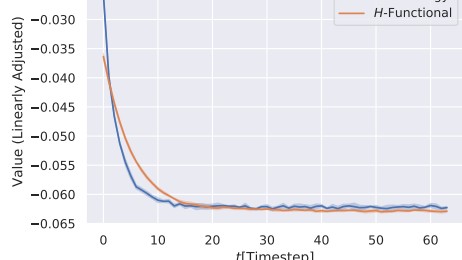

Figure 6: The *true* arrow of time (the Free-Energy functional, in blue) plotted against the learned arrow of time (the $H$-functional, i.e. the negative spatial expectation of the $h$-potential; plotted in orange) after linear scaling and shifting. **Gist:** we find the $H$-functional to be in good (albeit not perfect) agreement with the Free-Energy functional, where the latter is a known notion of an arrow of time.

where the first expectation of the RHS is the energy functional, and the second expectation is the negative entropy. A celebrated result due to Jordan, Kinderlehrer, and Otto (1998) is that the Free-Energy is a Lyapunov functional of the dynamics, i.e. it can only decrease with time, thereby defining a notion of an arrow of time. Now, to find out how well our learned arrow of time agrees with the Free-Energy functional, we train it with realizations

of the stochastic process $\mathbf{X}(t)$ in two-dimensions. Fig 6 plots the Free-Energy functional $F$ against a linearly adjusted $H$-functional, defined as: $H[\rho(\cdot, t)] = -\mathbb{E}_{\mathbf{x} \sim \rho(\cdot, t)}[h(\mathbf{x})]$. Indeed, we find that up to a linear transform, the $H$-functional (and the corresponding $h$-potential) agrees reasonably well with the true arrow of time given by the Free-Energy functional $F$. Crucially, the $H$-functional is also a Lyapunov functional of the dynamics – implying that in expectation over states, the $h$-potential functions as an arrow of time. Details can be found in App C.4.

## CONCLUSION

In this work, we approached the problem of learning an arrow of time in a Markov (Decision) Processes. We defined the arrow of time ($h$-potential) as a solution to an optimization problem and laid out the conceptual roadblocks that must be cleared before it can be useful in the RL context. But once these roadblocks have been cleared, we demonstrated how the notions of reachability, safety and curiosity can be bridged by a common framework of a learned arrow of time. Finally, we empirically investigated the strengths and shortcomings of our method on a selection of discrete and continuous environments. Future work could draw connections to algorithmic independence of cause and mechanism (Janzing et al., 2016) and explore applications in causal inference (Janzing, 2010; Peters et al., 2017).

## ACKNOWLEDGEMENTS

The authors would like to thank Min Lin for the initial discussions, Georgios Arvanitidis, Simon Ramstedt, Zaf Ahmed, Stefan Bauer and Maximilian Puelma Touzel for their valuable feedback on the draft.

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

# A    THEORETICAL ANALYSIS

In this section, **(a)** we present a theoretical analysis of the optimization problem formulated in Eqn 1, **(b)** analytically evaluate the result for a few toy Markov processes to validate that the resulting solutions are indeed consistent with intuition and (c) highlight how the $h$-potential differs from a forward state-action transition model. To simplify the exposition, we consider the discrete case where the state space $\mathcal{S}$ of the MDP is finite.

## A.1    ANALYTICAL SOLUTION TO EQN 1 AND APPLICATION TO TOY MARKOV CHAINS

Consider a discrete Markov chain with enumerable states $s_i \in \mathcal{S}$. At an arbitrary (but given) time-step $t$, we let $p_i^t = p(s_t = s_i)$ denote the probability that the Markov chain is in state $s_i$, and $\mathbf{p}^t$ the corresponding vector (over states). With $T_{ij}$ we denote the probability of the Markov chain transitioning from state $s_i$ to $s_j$ under some policy $\pi$, i.e. $T_{ij} = p_\pi(s_{t+1} = s_j | s_t = s_i)$. One has the transition rule:

$$\mathbf{p}^{t+1} = \mathbf{p}^t T \qquad \mathbf{p}^t = \mathbf{p}^0 T^t \tag{9}$$

where $T^t$ is the $t$-th matrix power of $T$. Now, we let $h_i$ denote the value $h_\pi$ takes at state $s_i$, i.e. $h_i = h_\pi(s_i)$, and the corresponding vector (over states) becomes $\mathbf{h}$. This reduces the expectation of the function (now a vector) $\mathbf{h}$ w.r.t any state distribution (now also a vector) $\mathbf{p}$ to the scalar product $\mathbf{p} \cdot \mathbf{h}$. In matrix notation, the optimization problem in Eqn 1 simplifies to:

$$\arg\max_{\mathbf{h}} \frac{1}{N} \sum_{t=0}^{N-1} \left[ \mathbf{p}^t T \mathbf{h} - \mathbf{p}^t \cdot \mathbf{h} \right] + \lambda \mathcal{T}(\mathbf{h}) \tag{10}$$

For certain $\mathcal{T}$, the discrete problem in Eqn 10 can be handled analytically. We consider two candidates for $\mathcal{T}$, the first being the norm of $\mathbf{h}$, and the second one being the norm of change in $h_i$ in expectation along trajectories.

**Proposition 1.** *If* $\mathcal{T}(\mathbf{h}) = -(2N)^{-1}\|\mathbf{h}\|^2$*, the solution to the optimization problem in Eqn 10 is given by:*

$$\mathbf{h} = \frac{\mathbf{p}^0 T^N - \mathbf{p}^0}{\lambda} \tag{11}$$

*Proof.* First, note that the objective in Eqn 10 becomes:

$$\mathcal{L}[\mathbf{h}] = \frac{1}{N} \sum_{t=0}^{N-1} \left[ \mathbf{p}^t T \mathbf{h} - \mathbf{p}^t \cdot \mathbf{h} \right] - \frac{1}{2N} \|\mathbf{h}\|^2 \tag{12}$$

To solve the maximization problem, we must differentiate $\mathcal{L}$ w.r.t. its argument $\mathbf{h}$, and set the resulting expression to zero. This yields:

$$\nabla_{\mathbf{h}} \mathcal{L} = \frac{1}{N} \left[ \sum_{t=0}^{N-1} (\mathbf{p}^t T - \mathbf{p}^t) - \lambda \mathbf{h} \right] = 0 \tag{13}$$

Now, the summation (over $t$) is telescoping, and evaluates to $\mathbf{p}^{N-1}T - \mathbf{p}^0$. Substituting $\mathbf{p}^{N-1}$ with the corresponding expression from Eqn 9 and solving for $\mathbf{h}$, we obtain Eqn 11.    □

Proposition 1 has an interesting implication: if the Markov chain is initialized at equilibrium, i.e. if $\mathbf{p}^0 = \mathbf{p}^0 T$, we obtain that $\mathbf{h} = \mathbf{0}$ identically. Given the above, we may now consider as examples the following simple Markov chains.

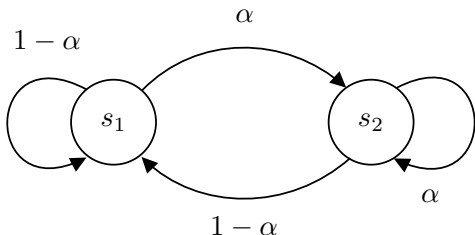

Figure 7: The two-state Markov chain considered in Examples 1 and 3.

**Example 1.** Consider a Markov chain with two states and reversible transitions, parameterized by $\alpha \in [0, 1]$ such that $T_{11} = T_{21} = 1 - \alpha$ and $T_{12} = T_{22} = \alpha$ (Fig 7). If $\mathbf{p}^0 = (1/2, 1/2)$, one obtains:

$$\mathbf{h} \propto (-\gamma, \gamma) \tag{14}$$

where $\gamma = \alpha - 1/2$. To see how, consider that for all $N > 0$, one obtains $\mathbf{p}^0 T^N = (1 - \alpha, \alpha)$. Together with Proposition 1, Eqn 14 follows.

The above example illustrates two things. On the one hand, if $\alpha = 1/2$, one obtains a Markov chain with *perfect reversibility*, i.e. the transition $s_1 \to s_2$ is equally as likely as the transition $s_2 \to s_1$. In this case, one indeed obtains $h(s_1) = h(s_2) = 0$, as mentioned above. On the other hand, if one sets $\alpha = 1$, the transition from $s_2 \to s_1$ is never sampled, and that from $s_1 \to s_2$ is irreversible; consequently, $h(s_2) - h(s_1)$ takes the largest value possible. Now, while this aligns well with our intuition, the following example exposes a weakness of the L2-norm-penalty used in Proposition 1.

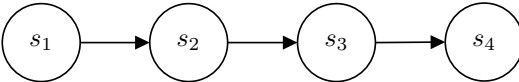

Figure 8: The four-state Markov chain considered in Examples 2 and 4.

**Example 2.** Consider two Markov chains, both always initialized at $s_1$. For the first Markov chain, the dynamics admits the following transitions: $s_1 \to s_2 \to s_3 \to s_4$, whereas for the second chain, one has $s_1 \to s_3 \to s_2 \to s_4$ (Fig 8). Now, for both chains and $N \geq 4$, it's easy to see that $(\mathbf{p}^0 T^N)_i = 1$ if $i = 4$, but 0 otherwise. From Eqn 11, one obtains:

$$\mathbf{h} \propto (-1, 0, 0, 1) \tag{15}$$

The solution for $h$ given by Eqn 15 indeed increases (non-strictly) monotonously with timestep. However, we obtain $h(s_2) = h(s_3) = 0$ for both Markov chains. In particular, $h$ does not increase between the $s_2 \to s_3$ transition in the former and the $s_3 \to s_2$ transition in the latter, even though both transitions are irreversible. It is in general apparent from 1 that the solution for $h$ depends only on the initial and final state distribution, and not the intermediate trajectory.

Now, consider the following regularizer that penalizes not just the function norm, but the change in $h$ in expectation along trajectories:

$$\mathcal{T}(\mathbf{h}) = -\frac{1}{2N} \sum_{t=0}^{N-1} (\mathbf{p}^t T \mathbf{h} - \mathbf{p}^t \cdot \mathbf{h})^2 - \frac{\omega}{2N} \|\mathbf{h}\|^2 \tag{16}$$

where $\omega$ is the relative weight of the L2 regularizer. This leads to the result:

**Proposition 2.** *The solution to the optimization problem in Eqn 10 with the regularizer in Eqn 16 is the solution to the following matrix-equation:*

$$\sum_{t=0}^{N-1} \mathbf{p}^0 (T^{t+1} - T^t) \mathbf{h} \, \mathbf{p}^0 (T^{t+1} - T^t) + \omega \mathbf{h} = \frac{\mathbf{p}^0 T^N - \mathbf{p}^0}{2\lambda} \tag{17}$$

*Proof.* Analogous to Eqn 12, we may write the objective in Eqn 10 as (by substituting Eqn 16 in Eqn 10):

$$\mathcal{L}[\mathbf{h}] = \frac{1}{N} \sum_{t=0}^{N-1} \left[\mathbf{p}^t T \mathbf{h} - \mathbf{p}^t \cdot \mathbf{h}\right] - \frac{\lambda}{2N} \sum_{t=0}^{N-1} (\mathbf{p}^t T \mathbf{h} - \mathbf{p}^t \cdot \mathbf{h})^2 - \frac{\lambda \omega}{2N} \|\mathbf{h}\|^2 \tag{18}$$

Like in Proposition 1, we maximize it by setting the gradient of $\mathcal{L}$ w.r.t. $\mathbf{h}$ to zero. This yields:

$$\nabla_{\mathbf{h}}\mathcal{L} = \frac{1}{N}\left[\sum_{t=0}^{N-1}(\mathbf{p}^t T - \mathbf{p}^t) - \frac{\lambda}{2}\nabla_{\mathbf{h}}\sum_{t=0}^{N-1}(\mathbf{p}^t T \mathbf{h} - \mathbf{p}^t \cdot \mathbf{h})^2 - \omega\lambda\mathbf{h}\right] = 0 \qquad (19)$$

The first term in the RHS is again a telescoping sum; it evaluates to: $\mathbf{p}^0 T^N - \mathbf{p}^0$ (cf. proof of Proposition 1). The second term can be expressed as (with $I$ as the identity matrix):

$$\frac{\lambda}{2}\nabla_{\mathbf{h}}\sum_{t=0}^{N-1}(\mathbf{p}^t T \mathbf{h} - \mathbf{p}^t \cdot \mathbf{h})^2 = \frac{\lambda}{2}\sum_{t=0}^{N-1}\nabla_{\mathbf{h}}(\mathbf{p}^t(T-I)\mathbf{h})^2 \qquad (20)$$

$$= \lambda\sum_{t=0}^{N-1}(\mathbf{p}^t(T-I)\mathbf{h})(\mathbf{p}^t(T-I)) \qquad (21)$$

$$= \lambda\sum_{t=0}^{N-1}\mathbf{p}^0(T^{t+1}-T^t)\mathbf{h}\,\mathbf{p}^0(T^{t+1}-T^t) \qquad (22)$$

where the last equality follows from Eqn 9. Substituting the above in Eqn 19 and rearranging terms yields Eqn 17. $\qquad\square$

While Eqn 17 does not yield an explicit expression for $\mathbf{h}$, it is sufficient for analysing individual cases considered in Examples 1 and 2.

**Example 3.** Consider the two-state Markov chain in Example 1 (Fig 7) and the associated transition matrix $T$ and initial state distribution $\mathbf{p}^0 = (1/2, 1/2)$. Using the regularization scheme in Eqn 16 and the associated solution Eqn 17, one obtains:

$$\mathbf{h} = (-\tilde{\gamma}, \tilde{\gamma}) \qquad (23)$$

where:

$$\tilde{\gamma} = \frac{2\alpha - 1}{\lambda(4\alpha^2 - 4\alpha + 2\omega + 1)} \qquad (24)$$

To obtain this result[7], we use that $T^t = T$ for all $t \geq 1$ and truncate the sum without loss of generality at $N = 1$.

Like in Example 1, we observe $h(s_1) = h(s_2) = 0$ if $\alpha = 1/2$ for all $\omega > 0$ (i.e. at equilibrium). In addition, if $\omega \geq 1/2$, it can be shown that $h(s_2) - h(s_1)$ increases monotonously with $\alpha$ and takes the largest possible value at $\alpha = 1$. We therefore find that for the simple two-state Markov chain of Example 1, the regularization in Eqn 16 indeed leads to intuitive behaviour for the respective solution $\mathbf{h}$. Now:

**Example 4.** Consider the four-state Markov chain with transitions $s_1 \to s_2 \to s_3 \to s_4$ (Fig 8) and the corresponding transition matrix $T$, where $T_{12} = T_{23} = T_{34} = T_{44} = 1$, $T_{ij} = 0$ for all other $i, j$. Set $\mathbf{p}^0 = (1, 0, 0, 0)$, i.e. the chain is always initialized at $s_1$. Now, the summation over $t$ in Eqn 17 can be truncated at $N = 4$ without loss of generality (over $N$), given that $T^{t+1} = T^t$ for all $t \geq 3$. At $\omega = 0$, one solution is:

$$\mathbf{h} \propto (-3/2, -1/2, 1/2, 3/2) \qquad (25)$$

Further, for all $\omega \geq 0$, one obtains $h(s_1) < h(s_2) < h(s_3) < h(s_4)$, where the inequality is strict. This is unlike Eqn 15 where $h(s_2) = h(s_3)$, and consistent with the intuitive expectation that the arrow of time must increase along irreversible transitions.

## A.2 THE $h$-POTENTIAL VS. A FORWARD MODEL

Thus far, we have considered Markov chains, which relies on a notion of a transition matrix $T_{ij}$ specifying $p(s_{t+1} = s_j | s_t = s_i)$. Now, the transition probabilities can also be expressed as:

$$p(s_j|s_i) := p(s_{t+1} = s_j | s_t = s_i) = \sum_a p(s_{t+1} = s_j | s_t = s_i, a_t = a)\pi(a|s_i) \qquad (26)$$

---

[7]Interested readers may refer to the attached SymPy computation.

where the variable $a$ is called the action, and $p(s_{t+1} = s_j | s_t = s_i, a_t = a)$ is the action-conditioned one-step forward transition model, or simply a forward model. The distribution $\pi(a|s_i)$ is called the policy, and can characterize the behaviour of an agent.

Now, given the forward model and a policy, one could define a possible measure of reversibility as:

$$g(s_i \to s_j) = \log \left[ \frac{p(s_j|s_i)}{p(s_i|s_j)} \right] \tag{27}$$

Indeed, $g(s_i \to s_j) = 0$ when $p(s_j|s_i) = p(s_i|s_j)$, i.e. when the probability of transitioning from state $s_i$ to state $s_j$ equals that of transitioning from $s_j$ to $s_i$. Further, if the transition $s_i \to s_j$ is more likely than $s_j \to s_i$ under the model and the policy, then we have that $p(s_j|s_i) > p(s_i|s_j)$ and consequently, $g(s_i \to s_j) > 0$ (and vice versa with $i$ and $j$ swapped). This raises the question: can the quantity $g(s_i \to s_j)$ replace $h(s_i \to s_j) := h(s_j) - h(s_i)$? To answer this, consider the following.

**First,** in non-ergodic processes, there may exist states $s_i$ and $s_j$ for which *both* quantities $p(s_j|s_i)$ and $p(s_i|s_j)$ are zero. In the Markov process in Figure 8, these could be states $(s_i, s_j) = (s_1, s_3)$ or $(s_i, s_j) = (s_1, s_4)$. In both cases, however, we have that $g(s_i \to s_j)$ is not defined. In fact, this applies to any functional form $g$ might take (i.e. it need not take the one specified in Eqn 27): as long as $g$ depends exclusively on $p(s_i|s_j)$ and $p(s_j|s_i)$, it is unable to differentiate between the two cases. This is quite unlike $h$, where we know from Example 4 that $h(s_1 \to s_3) = 2$ and $h(s_1 \to s_4) = 3$.

**Second,** to obtain the quantity $p(s_j|s_i)$ required to evaluate $g$, we require a marginalization over actions $a$. If $a$ is discrete and the action space is small, this is a simple summation. However, for large or even continuous action spaces, this marginalization amounts to an integral, which may not be tractable in practice.

It is therefore evident that the reversibility measure $g$ obtained with a one-step forward model need not be consistent over multiple steps, and in that it differs from the $h$-potential. One may address this by considering (in addition) $\tau$-step models $p(s_{t+\tau}|s_t, a_t, a_{t+1}, ..., a_{t+\tau-1})$, but to obtain $p(s_{t+\tau}|s_t)$ one must marginalize over $a_t, a_{t+1}, ..., a_{t+\tau-1}$, which does not scale well with $\tau$. Nevertheless, in practice it might be possible to utilize the one-step model as a mean to obtain the $h$-potential. This involves approximating the true transition matrix $T$ with a learned matrix $\tilde{T}$, which can then be used to analytically evaluate $h$ or to train a parameteric approximation to $h$ from trajectories sampled from the model in a manner analogous to DynaQ (Sutton & Barto, 2011).

CONCLUSION

In conclusion, we find that the functional objective defined in Eqn 1 may indeed lead to analytical solutions that are consistent with the notion of an arrow of time in certain toy Markov chains, and highlight the subtleties involved in relying on a one-step forward model to obtain a measure of reversibility. However, in most interesting real world environments, the transition model $T$ is not known and or or the number of states is infeasibly large, rendering an analytic solution intractable. In such cases, as we see in Section 5, it is possible to parameterize $h$ as a neural network and train the resulting model with stochastic gradient descent to optimize the functional objective defined in Eqn 1.

# B  ALGORITHM

---

**Algorithm 1** Training the $h$-Potential

---

**Require:** Environment `Env`, random policy $\pi_\sharp$, trajectory buffer `B`
**Require:** Model $h^\theta$, regularizer $\mathcal{T}$, optimizer.
 1: **for** $k = 1...M$ **do**
 2:   `B[k, :]` $\leftarrow (s_0, ..., s_N) \sim$ `Env[`$\pi_\sharp$`]` {Sample a trajectory of length $N$ with the random policy and write to $k$-th position in the buffer.}
 3: **end for**
 4: **loop**
 5:   Sample trajectory index $k \sim \{1, ..., M\}$ and time-step $t \sim \{0, ..., N-1\}$. {In general, one may sample multiple $k$'s and $t$'s for a larger mini-batch.}
 6:   Fetch states $s_t \leftarrow$ `B[k, t]` and $s_{t+1} \leftarrow$ `B[k, t + 1]` from buffer.
 7:   Compute loss as $L(\theta) = -[h^\theta(s_{t+1}) - h^\theta(s_t)]$.
 8:   **if** using trajectory regularizer **then**
 9:     Compute regularizer term as $[h^\theta(s_{t+1}) - h^\theta(s_t)]^2$ and add to $L(\theta)$.
10:   **else**
11:     Apply the regularizer as required. If early-stopping, break out of the loop if necessary.
12:   **end if**
13:   Compute parameter gradients $\nabla_\theta L(\theta)$ and update parameters with the optimizer.
14: **end loop**

---

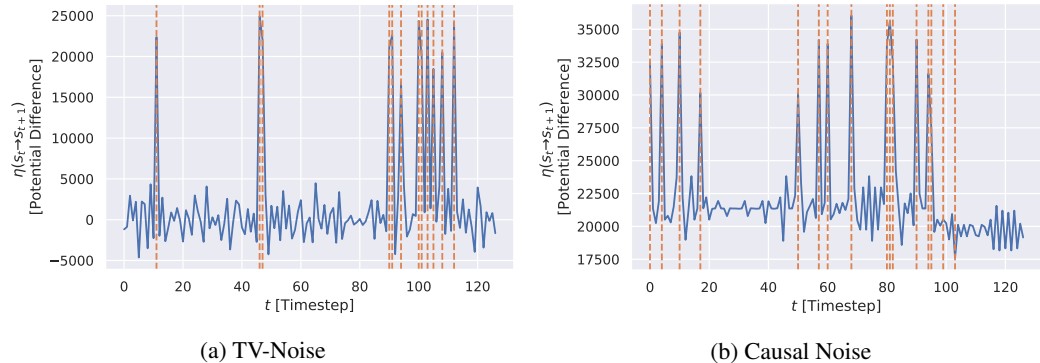

(a) TV-Noise                          (b) Causal Noise

Figure 9: The potential difference $\eta$ plotted along trajectories, where the state-space is augmented with temporally uncorrelated (TV-) and correlated (causal) noise. The dashed vertical lines indicate time-steps where a vase is broken. **Gist:** while our method is fairly robust to TV-noise, it might get distracted by causal noise.

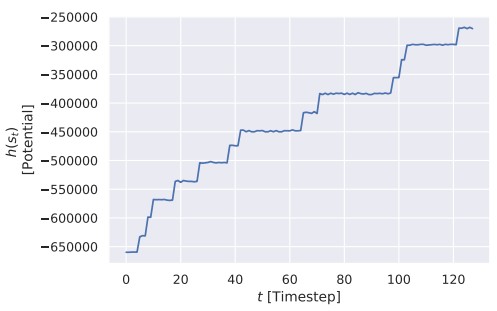

Figure 10: The $h$-potential along a trajectory sampled from a random policy. **Gist:** The $h$-potential increases step-wise along the trajectory every time an agent (irreversibly) breaks a vase. It remains constant as the agent (reversibly) moves about.

Figure 11: Histogram (over trajectories) of values taken by $h$ at time-steps $t = 0$, $t = 32$ and $t = T = 128$.

## C    EXPERIMENTAL DETAILS

All experiments were run on a workstation with 40 cores, 256 GB RAM and 2 nVidia GTX 1080Ti.

### C.1    DISCRETE ENVIRONMENTS

#### C.1.1    2D WORLD WITH VASES

The environment state comprises three $7 \times 7$ binary images (corresponding to agent, vases and goal), and the vases appear in a different arrangement every time the environment is reset. The probability of sampling a vase at any given position is set to ½.

We use a two-layer deep and 256-unit wide ReLU network to parameterize the $h$-potential. It is trained on 4096 trajectories of length 128 for 10000 iterations of stochastic gradient descent with Adam optimizer (learning rate: 0.0001). The batch-size is set to 128, and we use a weight decay of 0.005 to regularize the model. We use a validation trajectory to generate the plots in Fig 10 and 2. Moreover, Fig 11 shows histograms of the values taken by $h$ at various time-steps along the trajectory. We learn that $h$ takes on larger values (on average) as $t$ increases.

To test the robustness of our method, we conduct experiments where the environment state is augmented with one of: (a) a $7 \times 7$ image with uniform-randomly sampled pixel values (*TV-noise*) and (b) a $7 \times 7$ image where every pixel takes the value $t/128$, where $t$ is the time-step[8] of the correspond-

---

[8]Recall that the trajectory length is set to 128.

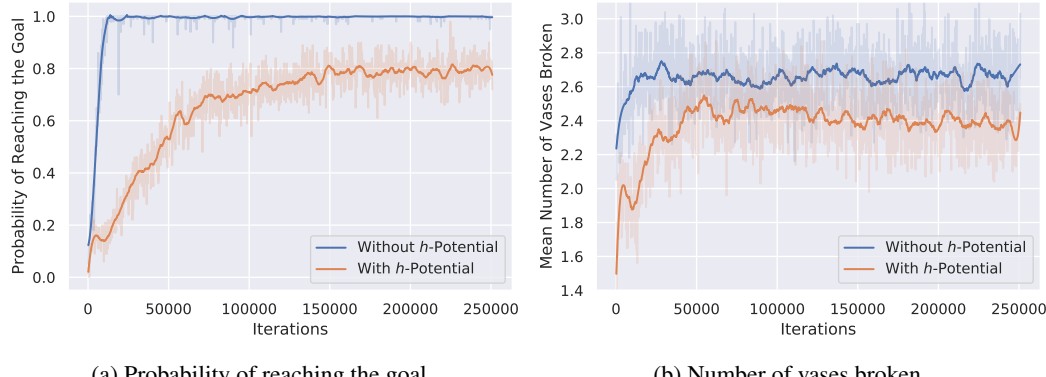

(a) Probability of reaching the goal.      (b) Number of vases broken.

Figure 12: Probability of reaching the goal and the expected number of vases broken, obtained over 100 evaluation episodes (per step). **Gist:** while the safety Lagrangian results in fewer vases broken, the probability of reaching the goal state is compromised. This trade-off between safety and efficiency is expected (cf. Moldovan & Abbeel (2012)).

ing state (*Causal Noise*). Fig 9a and 9b plot the corresponding $\eta = \Delta h$ along randomly sampled trajectories.

Given a learned arrow of time, we now present an experiment where we use it to derive a safe-exploration penalty (in addition to the environment reward). To that end, we now consider the situation where the agent's policy is not random, but specialized to reach the goal state (from its current state). For both the baseline and the safe agents, every action is rewarded with the change in Manhattan norm of the agent's position to that of the goal – i.e. an action that moves the agent closer to the goal is rewarded $+1$, one that moves it farther away from the goal is penalized $-1$, and one that keeps the distance unchanged is neither penalized nor rewarded ($0$). Further, every step is penalized by $-0.1$ (so as to keep the trajectories short), and exceeding the available time limit (30 steps) incurs a termination penalty ($-10$). In addition, the reward function of the safe agent is augmented with the reachability, i.e. it takes the form described in Eqn 4. We use $\beta = 4$ and a transfer function $\sigma$ such that $\sigma(\eta) = 0$ if $\eta < 5000$ (cf. Fig 2), and 1 otherwise.

The policy is parameterized by a 3-layer deep 256-unit wide (fully connected) ReLU network and trained via Duelling Double Deep Q-Learning[9] (Van Hasselt et al., 2016; Wang et al., 2015). The discount factor is set to 0.99 and the target network is updated once every 200 iterations. For exploration, we use a $1 - \epsilon$ greedy policy, where $\epsilon$ is decayed linearly from 1 to 0.1 in the span of the first 10000 iterations. The replay buffer stores 10000 experiences and the batch-size used is 10. Fig 12a shows the probability of reaching the goal (in an episode of 30 steps) over the iterations (sample size 100), whereas Fig 12b shows the expected number of vases broken per episode (over the same 100 episodes). Both curves are smoothed by a Savitzky-Golay filter (Savitzky & Golay, 1964) of order 3 and window-size 53 (the original, unsmoothed curves are shaded). As expected, we find that using the safety penalty does indeed result in fewer vases broken, but also makes the task of reaching the goal difficult (we do not ensure that the goal is reachable without breaking vases).

### C.1.2   2D World with Drying Tomatoes

The environment considered comprises a $7 \times 7$ 2D world where each cell is initially occupied by watered tomato plant[10]. The agent waters the cell it occupies, restoring the moisture level of the plant in the said cell to $100\%$. However, for each step the agent does not water a plant, it loses some moisture (by $2\%$ of maximum in our experiments). If a plant loses all moisture, it is considered dead and no amount of watering can resurrect it. The state-space comprises two $7 \times 7$ images: the first image is an indicator of the agent's position, whereas the pixel values of the second image quantifies the amount of moisture held by the plant[11] at the corresponding location.

---

[9]We adapt the implementation due to Shangtong (2018).

[10]We draw inspiration from the tomato-watering environment described in Leike et al. (2017).

[11]This is a strong causal signal which may distract the model. We include it nonetheless to make the task more challenging.

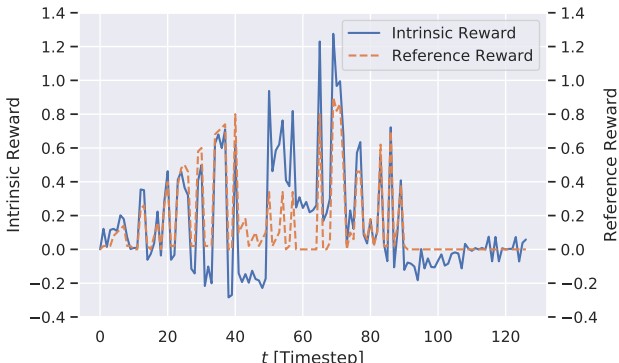

Figure 13: The intrinsic reward (Eqn 28) plotted against an engineered reward, which in this case is the amount of moisture gained by the tomato plant the agent just watered. **Gist:** the $h$-Potential captures useful information about the environment, which can then be utilized to define intrinsic rewards.

We show that it is possible to recover an intrinsic reward signal that coincides well with one that one might engineer. To that end, we parameterize the $h$-potential as a two-layer deep 256-unit wide ReLU network and train it on 4096 trajectories (generated by a random policy) of length 128 for 10000 iterations of Adam (learning rate: 0.0001). The batch-size is set to 128 and the model is regularized with the trajectory regularizer ($\lambda = 0.5$).

Unsurprisingly, we find that $h$ increases as the plants lose moisture. But conversely, when the agent waters a plant, it causes the $h$-potential to decrease by an amount that strongly correlates with the amount of moisture the watered plant gains. This can be used to define a *dense* reward signal for the agent:

$$\hat{r}_t = -\{\eta(s_{t-1} \to s_t) - \text{RunningAverage}_t[\eta]\} \tag{28}$$

where we use a momentum of $0.95$ to evaluate the running average.

In Fig 13, we plot for a random trajectory the intrinsic reward $\hat{r}_t$ against a reference reward, which in this case is the moisture gain of the plant the agent just watered. Further, we observe the reward function dropping significantly at around the 90-th iteration - this is precisely when all plants have died. This demonstrates that the $h$-potential can indeed be useful for defining intrinsic rewards.

### C.1.3 SOKOBAN

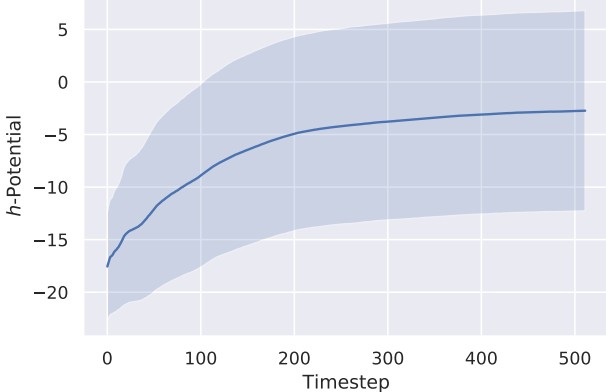

Figure 14: $h$-Potential averaged over 8000 trajectories, plotted against timestep $t$; shaded band shows the standard deviation. **Gist:** as required by its objective (Eqn 1), the $h$-Potential must increase in expectation along trajectories.

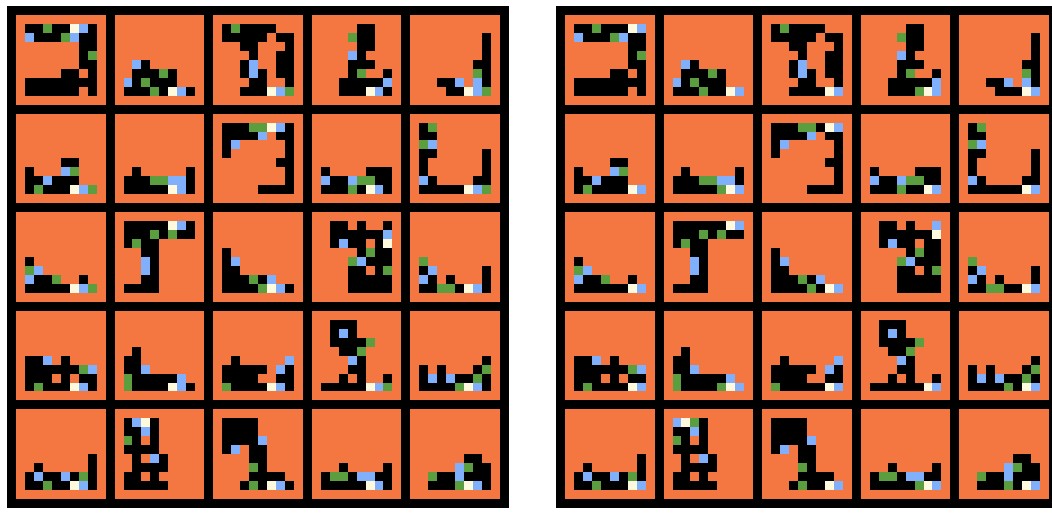

(a) States at time $t$.          (b) States at time $t+1$.

Figure 15: Random samples from 200 transitions that cause the largest increase in the $h$-potential (out of a sample size of 8000 transitions). The orange, white, blue and green sprites correspond to a wall, the agent, a box and a goal marker respectively. **Gist:** pushing boxes against the wall increases the $h$-potential.

The environment state comprises five $10 \times 10$ binary images, where the pixel value at each location indicates the presence of the agent, a box, a goal, a wall and empty space. The layout of all sprites are randomized at each environment reset, under the constraint that the game is still solvable (Schrader, 2018). The $h$-potential is parameterized by a two-layer deep and 512-unit wide network, which is trained on 4096 trajectories of length 512 for 20000 steps of Adam (learning rate: 0.0001). The batch-size is set to 256 and we use the trajectory regularizer ($\lambda = 0.05$) to regularize our model.

### C.1.4 Conveyor Belt Environment of Krakovna et al. (2018)

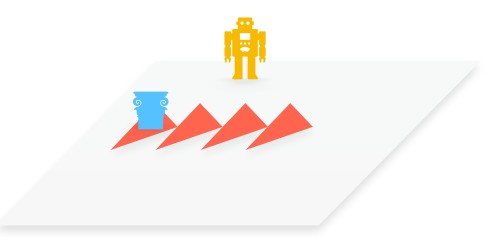

Figure 16: The initial state of the conveyor belt environment with an agent (orange robot), a vase (blue) and a conveyor belt (red arrows). The conveyor belt carries the vase rightwards, until it falls off it and breaks.

The environment considered is a $5 \times 5$ 2D world (Fig 16) with an agent, a conveyor belt and a vase that is initially placed on the belt (this is the *Vase environment* in Krakovna et al. (2018)). If the agent is *passive* (i.e. it stays put, Fig 17b), the conveyor belt moves the vase one step to the right until it eventually falls off the belt and breaks. However, the intended behaviour of a safe (*good*) agent (Fig 17a) is that it removes the vase from the conveyor belt, preventing it from breaking. One may also have a *malicious* agent (Fig 17c) that removes the vase from the belt (e.g. to collect a reward) only to put it back on it again. In contrast, an *inept* agent (Fig 17d) may remove the vase from the belt but irreversibly push it to a corner (like in Sokoban). In the following, we investigate the safety-reward that is awarded by the $h$-potential to each of these policies, which we label *good*, *passive*, *malicious* and *inept*.

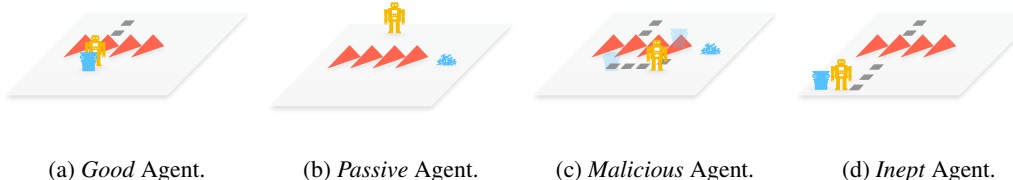

(a) *Good* Agent.     (b) *Passive* Agent.     (c) *Malicious* Agent.     (d) *Inept* Agent.

Figure 17: Illustration of the policies we use to test the safety reward assigned by the $h$-potential. **Gist:** the *good* agent removes the vase from the vase and stays put. The *passive* policy stays put and does nothing as the vase falls off the belt. The *malicious* policy removes the vase from the belt (possibly to collect a reward) only to put it back on it again. The *inept* policy removes the vase from the belt, but pushes it to a corner (the agent lacks the ability to pull it back).

To that end, we gather $4096$ random trajectories of length $64$ each. The $h$-potential is parameterized by a 256-unit wide and 2-layer deep MLP with ReLU activation, and trained with 80000 steps of stochastic gradient descent with Adam Kingma & Ba (2014). We use a trajectory regularizer with $\lambda = 0.01$ to regularize the model. The state space is a collection of 6 binary images of size $7 \times 7$, where in each image the truth value of a pixel marks the presence of a wall, empty space, belt, agent, vase and a broken vase in the corresponding location[12].

As the model trains, we track the return it awards to the four policies mentioned above (to aid visualization, we normalize the rewards to have a mean of zero) and plot the result in Fig 18. We find that the $h$-potential rewards the *good* policy, but penalizes the *passive*, *malicious* and *inept* policies. This can be contrasted with the predefined safety performance measure, which assigns a safety score of $+50$ to the *good* and *inept* agents, $0$ to the malicious agent and $-50$ to the passive agent[13] (larger score is safer). Our method therefore learns that pushing the vase to a corner is no less irreversible than breaking it (*malicious* and *passive*) and penalizes the *inept* policy accordingly.

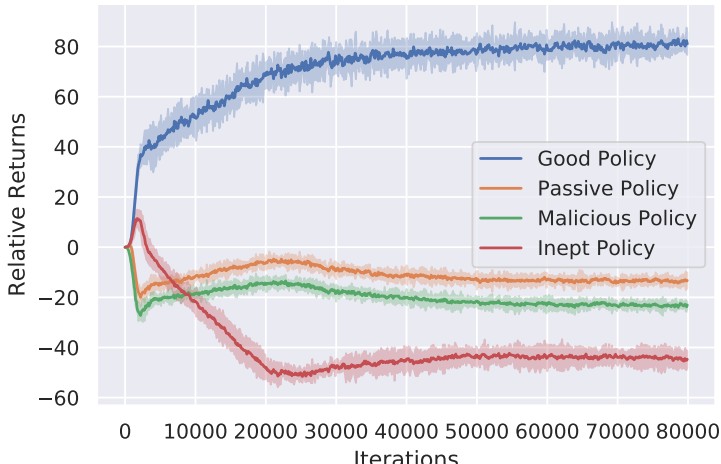

Figure 18: The normalized returns awarded to the various policies discussed in text and illustrated in Fig 17, averaged over 5 training runs (shaded bands are the standard deviations). **Gist:** the h-potential penalizes all irreversible behaviour, including the vase being pushed in to a corner by the *inept* policy (in addition to it falling off the belt due to the *passive* and *malicious* policies).

To conclude, we confirm that the safety reward extracted from the $h$-potential can enable agents to avoid irreversible behaviour. However, while preventing the irreversible is safe in this context, it may not always be the case – we point the reader to the Sushi environment in Krakovna et al. (2018) for an example[14].

---

[12]cf. implementation in `https://github.com/deepmind/ai-safety-gridworlds` under `environments/conveyor_belt.py`.

[13]We refer to the implementation in `https://github.com/deepmind/ai-safety-gridworlds`

[14]The object on the belt is Sushi instead of a vase, and the belt leads to a hungry human.

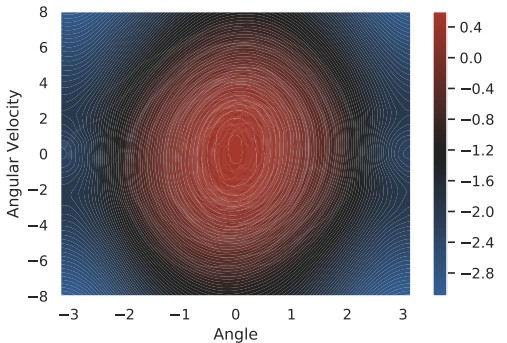

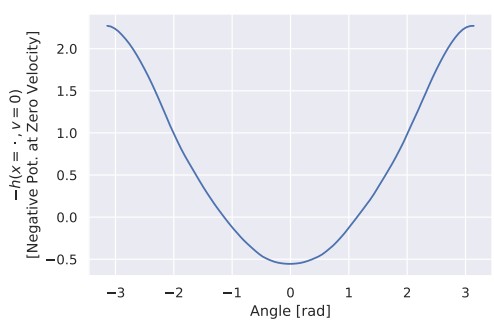

(a) Learned $h$-Potential as a function of the state-space $(\theta, \dot{\theta})$. Overlaid are trajectories from a random policy.

(b) Negative of the learned $h$-Potential as a function of $\theta$ when $\dot{\theta} = 0$.

Figure 19: **Gist:** the learned $h$-Potential takes large values around $(\theta, \dot{\theta}) = \mathbf{0}$, since that is where most trajectories terminate due to the effect of damping.

## C.2 CONTINUOUS ENVIRONMENTS

### C.2.1 UNDER-DAMPED PENDULUM

**Under-damped Pendulum.** The environment considered simulates an under-damped pendulum, where the state space comprises the angle[15] $\theta$ and angular velocity $\dot{\theta}$ of the pendulum. The dynamics are governed by the following differential equation where $\tau$ is the (time-dependent) torque applied by the agent and $m$, $l$, $g$ are constants:

$$\ddot{\theta} = \frac{-3g}{2l}\sin(\theta) + \frac{3\tau}{ml^2} - \alpha\dot{\theta} \tag{29}$$

We adapt the implementation in OpenAI Gym (Brockman et al., 2016) to add an extra term $\alpha\dot{\theta}$ to the dynamics to simulate friction. In our experiments, we set $g = 10$, $m = l = 1$, $\alpha = 0.1$ and the torque $\tau$ is uniformly sampled iid. from the interval $[-2, 2]$.

The $h$-Potential is parameterized by a two-layer 256-unit wide ReLU network, which is trained on 4096 trajectories of length 256 for 20000 steps of stochastic gradient descent with Adam (learning rate: 0.0001). The batch-size is set to 1024 and we use the trajectory regularizer with $\lambda = 1$ to regularize the network. Fig 19a plots the learned $h$-potential (trained with trajectory regularizer) as a function of the state $(\theta, \dot{\theta})$ whereas Fig 19b shows the negative potential for all angles $\theta$ at zero angular velocity, i.e. $\dot{\theta} = 0$. We indeed find that states in the vicinity of $\theta = 0$ have a larger $h$-potential, owing to the fact that all trajectories converge to $(\theta, \dot{\theta}) = \mathbf{0}$ for large $t$ due to the dissipative action of friction.

### C.2.2 CONTINUOUS MOUNTAIN CAR

The environment[16] considered is a variation of Mountain Car (Sutton & Barto, 2011), where the state-space is a tuple $(x, \dot{x})$ of the position and velocity of a vehicle on a mountainous terrain. The action space is the interval $[-1, 1]$ and denotes the *force f* applied by the vehicle. The dynamics of the modified environment is given by the following equation of motion:

$$\ddot{x} = \zeta f - 0.0025\cos 3x - \alpha\dot{x} \tag{30}$$

where $\zeta$ and $\alpha$ are constants set to $0.0015$ and $0.1$ respectively, and the velocity $\dot{x}$ is clamped to the interval $[-0.07, 0.07]$. Our modification is the last $\alpha\dot{x}$ term to simulate friction. Further, the initial state $(x, \dot{x})$ is sampled uniformly from the state space $\mathcal{S} = [-1.2, 0.6] \times [-0.07, 0.07]$. This

---

[15] $\theta$ is commonly represented as $(\cos(\theta), \sin(\theta))$ instead of a scalar.

[16] We adapt the implemetation due to Brockman et al. (2016), available here: github.com/openai/gym/blob/master/gym/envs/classic_control/continuous_mountain_car.py

can potentially be avoided if an exploratory policy is used (instead of the random policy) to gather trajectories, but we leave this for future work.

The $h$-potential is parameterized by a two-layer 256-unit wide ReLU network, which is trained on 4096 trajectories of length 256 for 20000 steps of stochastic gradient descent with Adam (learning rate: 0.0001). The batch-size is set to 1024 and we use the trajectory regularizer with $\lambda = 1$.

### C.3 LEARNING THE $h$-POTENTIAL WHILE SIMULTANEOUSLY EXPLORING THE ENVIRONMENT

Recall that the policy we have thus-far used to gather the trajectories required to train the $h$-potential is random (cf. Sec 2.2). While the use of random policies is ubiquitous in the model-based and related literature (Ha & Schmidhuber, 2018; Savinov et al., 2018; Kulkarni et al., 2019; Anand et al., 2019), it typically comes at a price: namely, the lack of adequate exploration in complex enough environments (Ha & Schmidhuber, 2018). In this section, we investigate possible strategies towards approaching this problem in the context of learning an $h$-potential. We stress that the results in this section are preliminary and the discussion below is intended to showcase the challenges that lie in the way; much more future work will be needed to holistically address this important issue in a principled manner.

To proceed, we consider again the environment of Mountain Car with Damping (cf. Sec 5 and App C.2.2), but with the amendment that the car is initialized in the valley with zero velocity. We choose this task because it is *small* enough for fast iteration and easy visualization, yet it poses a difficult exploration problem for an appropriate choice of environment parameters[17], given that the car may not climb far out of the valley by simply applying a constant action. Likewise, random actions are not enough to adequately explore the state-space in order to learn the $h$-potential (Fig 20).

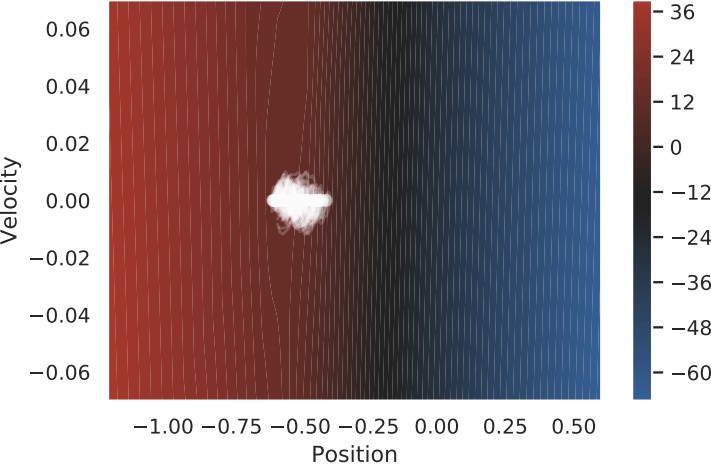

Figure 20: Random trajectories (white curves emanating from the dots) overlaid on the $h$-potential as a function of state (position and velocity). **Gist:** the $h$-potential trained on random trajectories fails to wholly characterize the dynamics of the considered environment. This is due to a lack of adequate exploration by the random policy.

The approach we describe can be thought of as *bootstrapping* the $h$-potential by using an exploratory policy in tandem with a random policy and a trajectory buffer. The procedure (adapted from Anonymous (2019)) is as following:

1. Initialize a trajectory buffer and fill it with trajectories from a random policy (starting at the environment specified initial states, i.e. in the valley).

---

[17]In particular, the friction parameter $\alpha$ and the force $\zeta$ in Eqn 30, where the latter can be thought of as the *power* of the car. We use $\alpha = 0.1$ and $\zeta = 0.0025$, which we assure ourselves is not enough for the car to reach the top of the mountain by applying a constant action.

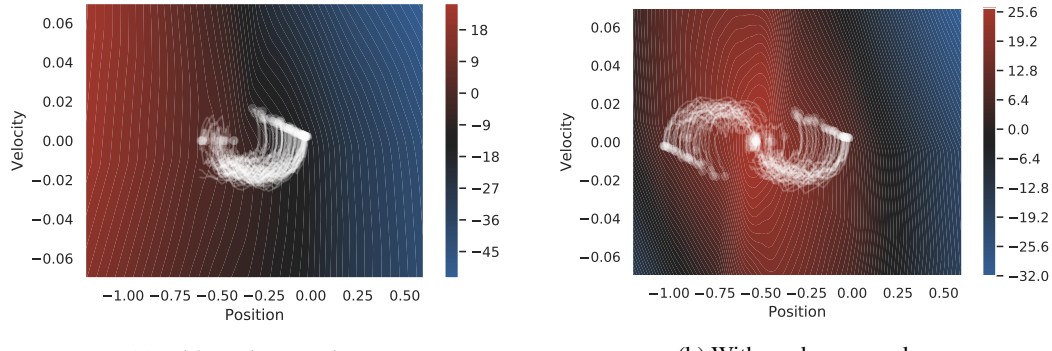

(a) With *exploration bias*.

(b) With random rewards.

Figure 21: The $h$-potential as a function of state (position and velocity). The overlaid curves (in white, emanating from the dots) show samples from the trajectory buffer used to train the respective $h$-potential. **Gist:** we find in Fig 21a that the exploration bias causes the $h$-potential to over-specialize to one section of the state-space whilst ignoring the other. This can be contrasted with Fig 21b, where the trajectories are gathered by initializing a random policy at states reached by exploratory policies trained to maximize random reward functions.

2. Train (from scratch) the $h$-potential with transitions available in the trajectory buffer.

3. Train (from scratch) an exploratory policy to minimize the $h$-potential (cf. Sec 3.3).

4. Use the exploratory policy to transition to a difficult-to-reach state.

5. Initialize the random policy at the said state, and use it to gather more trajectories that randomly replace a fraction[18] of the existing trajectories in the buffer.

6. Repeat steps 2-6.

The trajectories gathered in step 5 are mixed with previously gathered trajectories to improve the stability of the training procedure. Moreover, note that the $h$-potential and the exploratory policies are reinitialized and trained from scratch at every iteration (step 6) to counteract the *exploration-bias* detailed below. Further, the exploratory policy is parameterized by a NoisyNet (Fortunato et al., 2017) to (locally) aid exploration.

Unfortunately, the above procedure runs in to the problem that the exploratory policy adapts to the $h$-potential (in step 3), and the $h$-potential in turn adapts to the exploratory policy (in step 5). This circular adaptation, which we call the *exploration bias*, leads to the situation where the $h$-potential (correctly) learns that the mountain to the right is difficult to climb (Fig 21a, positions $> -0.52$), whereas the mountain to the right is left unexplored (because it is initially assigned a larger values). Consequently, the exploratory policy minimizing the $h$-potential will focus on climbing the mountain to the right whilst ignoring the one to the left (cf. Fig 21a), which in results in more trajectories gathered for the right mountain (and none for the left).

To side-step the *exploration bias*, we resort to pre-populating the trajectory buffer with trajectories gathered by a random policy initialized at states reached by exploratory policies trained on random reward potentials. Precisely, we replace the the the $h$-potential by a randomly initialized neural network, use it to train the exploratory policy, use the exploratory policy for a random number of steps to transition to a state, use that state as the initial state of a random policy, gather trajectories to populate the buffer, and repeat a few times over (5 in our experiments). The result is a trajectory buffer that is populated with a more diverse set of trajectories. The intuition behind this heuristic is that a random reward potential will attract the exploratory policy to random locations in the state space, leading to a more diverse set of trajectories. That said, we have not studied the efficiency of this strategy and its feasibility in other environments.

---

[18]0.5 in our experiments.

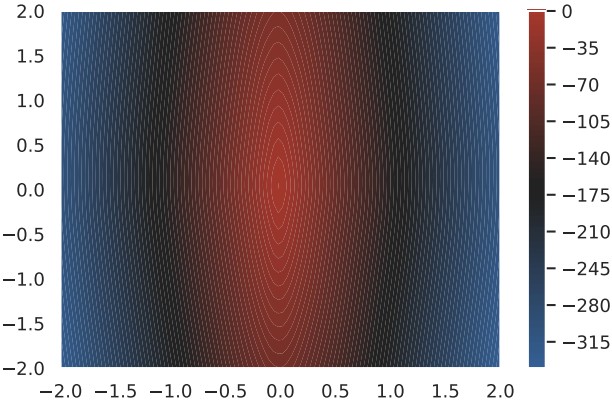

Figure 22: Learned $h$-Potential as a function of position $\mathbf{x}$. Observe the qualitative similarity to the potential $\Psi$ defined in Eqn 32.

With the trajectory buffer pre-populated, we train the $h$-potential to obtain the result[19] in Fig 21b, which can be compared to Fig 5a.

In conclusion, we presented a preliminary investigation of strategies towards attacking the exploration issues that may arise when training the $h$-potential (or model-based methods in general). Concretely, we formulated the strategy of bootstrapping (Anonymous, 2019), but found that it can be subject to what we called *exploration bias*. We side-stepped the *exploration bias* by using a population of exploratory policies trained to maximize random reward functions, which lead to trajectories diverse enough to train the $h$-potential for the environment considered. We remain optimistic that a well-crafted algorithm combining bootstrapping with random rewards might be fruitful even in complex environments, but leave a thorough investigation to future work.

### C.4 COMPARISON WITH THE FREE-ENERGY FUNCTIONAL

The environment state at a given time-step $t$ comprises two scalars, the $x_1(t)$ and $x_2(t)$ coordinates of the particle's position $\mathbf{x}(t)$. Recall that the dynamics is defined by:

$$d\mathbf{X}(t) = -\nabla\Psi(\mathbf{X}(t))dt + \sqrt{2\beta^{-1}}d\mathbf{W}(t) \tag{31}$$

where $\mathbf{X}(t)$ is the stochastic process associated with the particle's position $\mathbf{x}(t)$. In our experiments, the potential is given by:

$$\Psi(\mathbf{x}) = \frac{x_1^2}{20} + \frac{x_2^2}{40} \tag{32}$$

which makes $\mathbf{X}(t)$ a two dimensional Ornstein-Uhlenbeck process with temperature parameter $\sqrt{2\beta^{-1}}$ set to 0.3. Further, $\mathbb{E}_{\mathbf{x}\sim\rho(\cdot,t)}[\Psi]$ (in Eqn 8) is estimated via Monte-Carlo sampling, the differential entropy $\mathbb{E}_{\mathbf{x}\sim\rho(\cdot,t)}[\log\rho(\cdot,t)]$ via a non-parametric estimator (Kozachenko & Leonenko, 1987; Kraskov et al., 2004; Gao et al., 2015), and the linear transform coefficients for $H$ via linear regression.

We train a two-layer deep, 512-unit wide network on 8092 trajectories of length 64 for 20000 steps of stochastic gradient descent with Adam (learning rate: 0.0001). The batch-size is set to 1024 and the network is regularized by weight decay (with coefficient 0.0005). Fig 22 shows the learned $h$-potential as a function of position $\mathbf{x}$. Fig 6 compares the free-energy functional with the learnt arrow of time given by the linearly scaled $H$-functional. To obtain the linear scaling parameters for the $H$, we find parameters $w$ and $b$ such that $\sum_{t=0}^{N}(wH[\rho(\cdot,t)] + b - F[\rho(\cdot,t)])^2$ is minimized (constraining $w$ to be positive), i.e. by solving a linear regression problem. Finally, Fig 22 plots $h$ as a function of state $\mathbf{x} \in \mathbb{R}^2$, whereas Fig 6 shows that after appropriate (linear) scaling, the learned $H$ largely agrees with the true $F$.

---

[19]We tried refining the $h$-potential with a few iterations of bootstrapping, but that did not significantly change the result.

The linear adjustment is done to account for the arbitrary scaling of $H$ and $F$; the crucial detail is that $H$ is also a Lyapunov functional of the dynamics, i.e. it decreases montonously with time. This arbitrariness results from various aspects. First, observe that while the dynamics in Eqn 31 is invariant to a constant shift in potential $\Psi$, the Free Energy functional $F$ is not – from Eqn 8, we see that adding a constant to the potential $\Psi$ results in the same constant being added to $F$ (for all $t$). This justifies adding a constant shift to $H$ as appropriate. Moreover, the scale of $h$-potential is controlled by the regularizing coefficient $\lambda$, which is arbitrary with respect to the scale of the Free Energy functional $F$.

