# OpenReview forum: "Learning the Arrow of Time for Problems in Reinforcement Learning"
_ICLR.cc/2020/Conference — Accept (Poster)_

### Official Review · AnonReviewer1 · 2019-10-16
**Official Blind Review #1**

**Rating:** 8

**Review:**

This work proposes the h-potential, which is a solution to an objective that measures state-transition asymmetry in an MDP. Roughly speaking, in many situations some state transitions (s-->s’) are more probable than their converse (s’-->s), and if we have a function that assigns a higher value to a more probable transition (compared to its converse), then we can use it as a measure of the “reversibility” of that transition. This function can then be used, for example, as an intrinsic reward signal; indeed, there may be cases where state transitions should be avoided if they are not reversible.

The authors tie these ideas into the notion of the arrow of time. They go on to explore the various nuances and subtleties of this measure, and demonstrate its behaviour empirically on a number of environments.

This paper was an absolute pleasure to read. The prose was clear, interesting, and nuanced. The authors anticipate many questions and do well to explain the various subtleties of their method. Overall the experiments are a nice demonstration of the presented ideas.

I am inclined to give this paper a high rating, as there was very little that I felt was “wrong” or inaccurate. But I must also admit that some of the theoretical components are beyond my expertise, and so I can only be moderately confident. I will defer to other reviewers and any online discussion on these more technical matters.

I have a few questions that I hope the authors can address.

1) The method depends on a random policy (or, more accurately, was empirically validated mainly using a random policy, aside from some very simple environments as far as I can see). Can the authors comment on the usefulness of this method for a non-random policy in more complicated environments? Do they have any experimental results showing the effect of incorporating this into an agent also receiving (and learning from) exogenous rewards in an environment such as Sokoban (as it’s used here already) or Atari?

2) Related to the first point, how does this method scale to environments whose state-space can only be sparsely covered with a random policy? It seems in this case a task-relevant policy would be needed to explore more of the state-space, which would place pressure on the h-potential function approximator as it has to learn with sequentially correlated inputs. You can imagine something like an experience replay buffer being needed, but in any case, there are definitely unique challenges here not explored in the paper.

3) Can the authors comment on the notion of a function being statistically monotonic vs. deterministically so, and whether the arrow of time is classically considered the former or the latter? My reasoning for this question comes from a place where I’m questioning whether the motivation of the work can be simplified. The appeal to the arrow of time is nice and reads well, but it’s also the case that this work can simply be interpreted as “learned state transition reversibility”, with the links to the arrow of time being more of a point of discussion.


**Experience Assessment:**

I do not know much about this area.

**Review Assessment: Checking Correctness Of Derivations And Theory:**

I did not assess the derivations or theory.

**Review Assessment: Checking Correctness Of Experiments:**

I assessed the sensibility of the experiments.

**Review Assessment: Thoroughness In Paper Reading:**

I read the paper at least twice and used my best judgement in assessing the paper.

---

> ### Author Response · Authors · 2019-11-07
> **Thank you for your questions and the positive review!**
>
> Thank you for your positive review -- we are elated that you enjoyed reading our work!
>
> > I have a few questions that I hope the authors can address.
>
> Those are all good questions.
>
> Regarding (1) and (2): the issue of non-random policy for gathering trajectories is perhaps the most important issue we do not address in this work, for doing it full justice would detract significantly from the original goal of the paper.
>
> Nevertheless, one approach that we see being fruitful is that of off-policy learning [1], which applies to cases where the behaviour policy used to gather trajectories differs from the evaluation policy of interest. In our case, the behaviour policy could be any combination of an exploratory policy and a learning agent, whereas the evaluation policy is random (in order to not bias the h-potential). Importance sampling is one example of this scheme, but more sophisticated methods exist. As such, your assessment that "there are unique challenges here not explored in the paper" is spot on.
>
> Moreover, it might also be worthwhile to remark that the adoption of random rollouts is rather widespread in model-based (related) literature [3-8].
>
> Regarding question (3): Consider a stochastic process $X_t$ (i.e. a (time-)series of random variables). We say that a (deterministic) function $h$ is statistically monotonic increasing if the function $H(t) = \mathbb{E} [h(X_t) | h(X_{t - 1}), ..., h(X_1)]$ is (deterministically) monotonic increasing with t. In other words, only the expectation of $h$ (w.r.t. its argument random variable) must increase with time (but not $h$ itself). In technical jargon, $h(X_t)$ is sometimes called a submartingale [2].
>
> All that said, we agree with your interpretation of our work as "learning the state transition reversibility", but with a crucial detail -- namely that the learner has a specific architecture resembling a siamese network, i.e. its output is a difference of a function applied to its two inputs. This function turns out to be the h-potential: cf. p. 4 of our manuscript: "Instead, our proposal is to derive [...]" (et seq.) for a discussion about what we may gain by using this specific architecture.
>
> We hope to have answered your questions -- please let us know if not!
>
> [1] Munos et al. 2016, "Safe and Efficient Off-Policy Reinforcement Learning." https://arxiv.org/abs/1606.02647
>
> [2] https://en.wikipedia.org/wiki/Martingale_(probability_theory)#Submartingales,_supermartingales,_and_relationship_to_harmonic_functions
>
> [3] Savinov et al. 2018, "Semi-parametric Topological Memory for Navigation." https://arxiv.org/abs/1803.00653
>
> [4] Ha & Schmidhuber 2018, "World Models." https://arxiv.org/abs/1803.10122
>
> [5] Savinov et al. 2018, "Episodic Curiosity Through Reachability." https://arxiv.org/abs/1810.02274
>
> [6] Nagabandi et al. 2017, "Learning Image-Conditioned Dynamics Models for Control of Under-actuated Legged Millirobots." https://arxiv.org/abs/1711.05253
>
> [7] Kulkarni et al. 2019, "Unsupervised Learning of Object Keypoints for Perception and Control." https://arxiv.org/abs/1906.11883
>
> [8] Anand et al. 2019, "Unsupervised State Representation Learning in Atari." https://arxiv.org/abs/1906.08226

---

> > ### Comment · AnonReviewer1 · 2019-11-14
> > **Thanks**
> >
> > Thank you for your response to my review and to the other reviewers. I am satisfied with the quality of the work and my score will remain unchanged.

---

> > > ### Author Response · Authors · 2019-11-14
> > > **Thank you!**
> > >
> > > We thank you for reviewing our work!

---

### Official Review · AnonReviewer2 · 2019-10-23
**Official Blind Review #2**

**Rating:** 6

**Review:**

The paper draws on a wide range of ideas, and proposes novel perspectives on how these ideas might apply in RL.  In particular, the concept of reachability, reversability and dissipation are explored, with respect to properties of the underlying MDP that can be exploited.

I found many of the ideas thought-provoking. The paper is also well written and a pleasure to read.

But unfortunately the work falls short of its objective in the experiment section.  Not a single baseline is included.  I would expect to see comparison to a simple model-based method for all the experiments on p.7.   The main result for 7x7 2D world seems to be that the agent has learned to quantify irreversibility.  I would expect a simple statistical estimator over state transitions (using the same samples as the h-potential method) to be able to capture this as well.  The main result for Sokoban seems to be that the h-potential has detected side-effects of actions; again, why can’t a model estimator learn this?  Similarly in Mountain car, it seems possible to directly estimate the terrain from the data, without the h-potential.

As a more minor concern, the fact that the method uses a batch of uniformly random state transitions (as per Sec.4), rather than randomly sampled trajectories is a definite concerned with respect to real-world application.

Minor comments:
-	Top of p.3: Can you give some intuition for h(), e.g. relation to entropy over trajectory.
-	Bottom of p.3: you use a random policy to sample trajectories. Is this simple to implement? Can you just sample random actions at each state, or do you need to sample over the space of all trajectories?
-	Footnote 2, p.4.  It would be interesting to expand on this point.

==============
Update post-rebuttal:
Indeed, my concern is that a simple T(s,a,s') estimator, using the same samples, could infer the same characteristics in each of the experiments (number of broken vases; effect of box pushing; terrain), and could then be used for model-based planning.
But I do appreciate the insights provided by connecting these through the broader notion of arrow of time developed in this paper, and its connection to reachability and safety.   Therefore I am raising my score to weak accept.

**Experience Assessment:**

I have published in this field for several years.

**Review Assessment: Checking Correctness Of Derivations And Theory:**

I assessed the sensibility of the derivations and theory.

**Review Assessment: Checking Correctness Of Experiments:**

I carefully checked the experiments.

**Review Assessment: Thoroughness In Paper Reading:**

I read the paper thoroughly.

---

> ### Author Response · Authors · 2019-11-07
> **First Response & Request for Clarifications (Part 1/2)**
>
> Thank you for your review - we are glad that you enjoyed reading our paper and find many of our ideas thought provoking.
>
> > I would expect to see comparison to a simple model-based method [...]
>
> To recapitulate, we propose a method to quantify irreversibility in MDPs without having to learn the full dynamics model of the said MDP. To that end, we obtain a function $\eta$ of two states $s, s'$ such that $\eta(s, s') = h(s') - h(s)$ is a scalar that quantifies the reversibility of the transition $s \to s'$ (cf. Sec 3.1). In contrast, a forward model is typically a function of a state $s$ and an action that outputs (a distribution over) the next state $s'$, i.e. $p(s' | s, a)$.  While we do not see a straightforward way of extracting a baseline measure of reversibility from the environment model, we would be very receptive to your suggestions.
>
> It is also worth noting that $\eta(s, s')$ has the functional constraint that it must be expressible as the difference $h(s') - h(s)$ of h-potentials (akin to a siamese neural network). This constraint functions as an inductive bias and endows it with the ability to generalise beyond the transitions that are observed (cf. Sec 3.1, especially p. 5 "Third, $\eta$ allows for a soft measure of reachability [...]"). In general, this cannot be expected of an environment model.
>
> > [...] for 7x7 2D world [...] I would expect a simple statistical estimator over state transitions (using the same samples as the h-potential method) to be able to capture this as well.
>
> It would certainly be possible to hand-craft an estimator that captures irreversibility of state transitions; this might be a challenging endeavor (or not), depending on the environment. However, our goal is to _learn_ a model that can capture irreversibility. The experiment on 2D World with Vases (Fig 2 and App C.1.1) therefore serves the following purposes. (1) It reassures us and our audience that the h-potential indeed learns to count the number of broken vases (which is precisely what one would consider when designing a statistical estimator). (2) It serves as a sandbox to understand the strengths and limitations of our method -- for instance, we find that our method is fairly robust against random uncorrelated noise, but can be distracted by time-correlated noise. (3) It helps expose the the expected trade-off between safety and efficiency (Fig 12) -- while the baseline DDQN agent trained without the h-potential out-performs the safe agent trained with the h-potential as far as the probability of reaching the goal is concerned, the baseline agent breaks a larger number of vases (i.e. is less safe) than the safe agent to acheive its goal.
>
> > Sokoban seems to be that the h-potential has detected side-effects of actions; again, why can’t a model estimator learn this?
>
> We would appreciate it if you could clarify what "learning a model estimator" might mean in this context. Please correct us if we are wrong, but we interpret your question as meaning either "Why not learn a model to directly predict side-effects?" or "Why not directly predict whether a transition is irreversible?".
>
> For the former: we often do not have access to ground-truth labels for when a transition induces side-effects. But if we did, it might also make sense to use it directly. For the latter: our method is indeed based on predicting whether a transition is irreversible, but with a crucial inductive bias: the corresponding predictor must be a difference of two functions (akin to a siamese neural network). Please refer to p. 4 ("Instead, our proposal is to derive [...]" et seq.) for a discussion.
>
> > As a more minor concern, the fact that the method uses a batch of uniformly random state transitions (as per Sec.4), rather than randomly sampled trajectories is a definite concerned with respect to real-world application.
>
> This might be a misunderstanding -- by uniform random policy, we mean that the actions are sampled randomly, with no action preferred over the other. This is straightforward to implement for the environments we consider. We will clarify this in the next update.
>
> (Continued in Part 2.)

---

> > ### Author Response · Authors · 2019-11-07
> > **First Response & Request for Clarifications (Part 2/2)**
> >
> > (Continued from Part 1.)
> >
> > This post addresses your minor comments.
> >
> > > Top of p.3: Can you give some intuition for h(), e.g. relation to entropy over trajectory.
> >
> > Certainly. Consider the random variable $S_t$ corresponding to the state of the system after t time-steps (more precisely, $S_t$ is a stochastic process). The $h()$ function is such that the expectation of $h(S_t)$ over $S_t$ should increase with increasing t, making it an "arrow of time" (or a Lyapunov functional of the dynamics, in technical terms). Note that this does not mean for samples $s_t$ of the random variable $S_t$ that $h(s_t)$ must always increase. For instance, observe in Fig 3 that $h()$ can decrease with time (around $t = 75$); but $h(S_t)$ must increase in expectation over all trajectories, as seen in Fig 16 (Appendix).
> >
> > The empirical analogy with the thermodynamical entropy arises via the explict comparison with the so called Free-Energy functional of a random walk (cf. "Comparison with the Free-Energy Functional"). A well known result in variational Fokker-Planck is that under the right assumptions, the free energy (which comprises an energy and a negative entropy term) must monotonously decrease with time.
> >
> > > Bottom of p.3: you use a random policy to sample trajectories. Is this simple to implement? Can you just sample random actions at each state, or do you need to sample over the space of all trajectories?
> >
> > We sample trajectories with a policy that selects an action at random (without consulting the state). Sampling uniformly from the set of all trajectories is indeed very non-trivial, but it is not what we do (we justify this in p. 3, "As a compromise, we use [...]"). We will clarify this in the update.
> >
> > > Footnote 2, p.4.  It would be interesting to expand on this point.
> >
> > In inverse reinforcement learning (IRL), the goal is to learn the reward function that is optimized by an expert agent. The hope is that a policy trained on this learned reward will mimic the expert in order to successfully solve the task. Our proposal can be thought of as replacing the expert agent with a "dumb" one. Accordingly, we ask "what reward function does a random policy maximize?". In doing so, we do not capture the pecularities of the expert agent, but only of the underlying environment. In informal terms, we ask "what does the environment want to do if left to its own devices?" The resulting function is analogous to the h-potential (up to minor technicalities).

---

> ### Author Response · Authors · 2019-11-13
> **Update Notification**
>
> We would like to notify you that we have updated our manuscript. Among other things, the new revision is more clear about the use of a random policy. Please refer to the general comment for a change-log: https://openreview.net/forum?id=rylJkpEtwS&noteId=B1ej2GptoB
>
> We believe to have addressed your concerns. Should that not be the case, please let us know as soon as possible.

---

> ### Author Response · Authors · 2019-11-14
> **(Final) Request for Clarification**
>
> Dear Reviewer 2, our newest revision includes new results, analysis and improved text (please see our general comment for a change-log [A]). Given the rebuttal period ends soon, we would appreciate a response from you on the points we have raised (in particular, our request for clarifications). Thank you for your time.
>
> [A] https://openreview.net/forum?id=rylJkpEtwS&noteId=B1ej2GptoB

---

### Official Review · AnonReviewer3 · 2019-10-25
**Official Blind Review #3**

**Rating:** 6

**Review:**

This paper proposes that we learn the “arrow of time” for an MDP: that is, a function (called the h-potential) that tends to increase as the MDP steps forward. Such an arrow should automatically capture notions such as irreversibility, and so can be used to define a measure of reachability, which previous work has shown can be used to penalize the agent for causing negative side effects. In addition, it can be used as intrinsic motivation for the agent: in particular, the agent can be rewarded for trajectories that decrease the h-potential (i.e. are “like” going backwards in time, or reducing entropy), which is hard to do and should lead to interesting skills. They propose that we learn the arrow of time by optimizing a function to grow over time along trajectories take from a random policy. Experiments demonstrate that in simple environments the learned function has the properties we would expect it to given results from physics.

I am conflicted on this paper. I like the novelty of the suggestion; it is not something I would have expected to see in an ML paper. The discussion and experiments have convinced me that the idea is worth investigating: they show that the learned arrow of time approximately satisfies the properties we would expect, and demonstrate their two use cases in two simple environments: the intrinsic reward is used in a tomato-watering environment, while the side effect avoidance is shown in Sokoban (though the experiment only shows that the h-potential increases with irreversible actions -- it doesn’t actually use the h-potential to create an agent that reliably avoids irreversibly pushing boxes into corners). However, it’s not clear to me whether or not the method would scale to more complex environments, the current experiments are more like demonstrations (there are no baselines), and the paper is hard to understand without a background in physics. Overall, given the novelty of the suggestion, I lean towards a weak accept.

----

In your objective, you use an expectation over the timestep in the trajectory. Why not instead take an average over all the timesteps in the trajectory? This should be equivalent. (If trajectories are all the same length, you could also take a sum over the timesteps.) Similarly, in the algorithm, why do you sample from the dataset? It would likely be better to randomly shuffle the dataset (at the timestep level) once, and then iterate through the dataset computing gradient updates. (This is standard practice in supervised learning.)

----

When scaling the algorithm up to larger environments, you will likely run into the problem that uniform policies are often very bad at exploring the state space. Consider for example the Overcooked environment (https://bair.berkeley.edu/blog/2019/10/21/coordination/ ). The environment is not dissipative, so you’d expect the h-potential to be zero. However, uniform policies are extremely unlikely to ever deliver a soup (which requires several hierarchical actions), but sometimes will pick up an onion. If you don’t know about soup delivery, then picking up an onion looks irreversible, because there’s no way to get rid of it, and the h-potential will rise. I think it would significantly improve this paper to demonstrate a solution to this problem.

One possibility is to redefine the arrow of time to be with respect to some distribution over states: E_{s ~ p(s)} E_{a ~ Uniform(A)} E_{s’ ~ p(s’ | s, a)} [ h(s’) - h(s) | s ]. Initially, your distribution over states can be the initial state distribution. But then you can use the learned arrow of time as an intrinsic reward to find a policy that finds “interesting states”. In Overcooked, we would hope that this policy learns to pick up onions. Then, your new distribution over states can be the states reached in 0-20 timesteps when following the new policy, that is, you collect your dataset by running the “interesting” policy for some time, and then switch back to the uniformly random policy, and only use the states / actions collected during the uniform random policy as part of your dataset. Hopefully, this would include states where the onion is placed in a pot, and the h-potential would learn that placing an onion in a pot is “irreversible”. Then, another round of using the h-potential would lead to a policy that places onions in pots. Collecting data could then discover delivering soups, and so on.

----

In addition to experiments on larger environments, I would like to see better experiments for the two intended use cases. For example, can you use the learned arrow of time to solve the environments in (Krakovna et al), and how does it compare to relative reachability and its many variants? Similarly, how does your intrinsic reward compare to existing exploration methods (of which there are many, but consider count-based methods, curiosity (Pathak et al), random network distillation (Burda et al))?

----

(This section did not affect my assessment of the paper)

Have you considered finding theoretically what it means to take the difference between the h-potential of two states (i.e. your reachability measure)? I could imagine that the answer is something like “reachability(s, s’) is proportional to the log probability of reaching s’ from s when acting according to a uniformly random policy”. Perhaps this has to be normalized against the log probability of reaching other states from s. This would be very interesting as a potential definition of reachability.

----

There is a lot of jargon from physics that will not be familiar to the typical audience at ICLR (e.g. Hamiltonian, Liouville’s theorem, Maxwell’s demon, free energy functionals, etc.) I would recommend improving the clarity of the paper on this axis. There’s no particular need to name Maxwell’s demon prominently -- the exposition is sufficient by itself, perhaps Maxwell’s demon can be mentioned in a footnote. Consider adding a discussion of ergodicity (as applied to Markov chains / MDPs), which will be more familiar to the audience and serves a similar purpose as the discussion on Hamiltonian systems. Perhaps move the experiment with the free energy functional to the appendix, and move the experimental details for the other experiments from the appendix to the main paper.

**Experience Assessment:**

I have read many papers in this area.

**Review Assessment: Checking Correctness Of Derivations And Theory:**

I did not assess the derivations or theory.

**Review Assessment: Checking Correctness Of Experiments:**

I assessed the sensibility of the experiments.

**Review Assessment: Thoroughness In Paper Reading:**

I read the paper at least twice and used my best judgement in assessing the paper.

---

> ### Author Response · Authors · 2019-11-07
> **First Response (Part 1/2)**
>
> Thank you for the time you've invested in reviewing our work -- we are glad that you find our suggestion novel, and are very grateful for your suggestions. The following is a first response to your comments; our manuscript will be updated in the coming days to reflect your suggestions and we will notify you once the changes are in place.
>
> > the paper is hard to understand without a background in physics [...] There is a lot of jargon from physics that will not be familiar to the typical audience at ICLR
>
> In hindsight, we agree with your assessment that some aspects of the paper might be difficult to understand without a background in physics, and part of this indeed has to do with the jargon. We will try to address this where possible.
>
> That said, we view our work as an attempt to view problems in RL from the lens of ideas and concepts that are already well understood in statistical physics. For instance, the connections between Variational Fokker-Planck and Reinforcement Learning is less explored (the only relevant reference we found was [1]). Likewise, the notion of dissipativity has received little attention in modern reinforcement learning (but has been well studied in the context of dynamical systems). We hope that our empirical results (especially "Comparison with the Free-Energy Functional", p. 8 and "[...] the Importance of Dissipativity", p. 7) inspires more theoretical research in this direction. Indeed, the fields of unsupervised learning and deep learning theory has greatly benifited from such interdisciplinary endeavors (e.g. [2], Energy Based Models, [3], and more), and we wish the same for reinforcement learning.
>
> > In your objective, you use an expectation over the timestep in the trajectory. Why not instead take an average over all the timesteps in the trajectory? This should be equivalent.
>
> They are indeed equivalent (and a matter of notation).
>
> > Similarly, in the algorithm, why do you sample from the dataset? It would likely be better to randomly shuffle the dataset (at the timestep level) once, and then iterate through the dataset computing gradient updates.
>
> It's certainly possible to partition the training in epochs, as you suggest. We will investigate if this leads to gains in performance.
>
> > When scaling the algorithm up to larger environments, you will likely run into the problem that uniform policies are often very bad at exploring the state space.
>
> The issue you mention is exceedingly important, and doing it justice will require us to deviate significantly from the primary objective of this work. We hope to have been upfront about it (cf. top of p. 4: "The price we pay is the lack of adequate exploration in complex enough environments [...]" and footnote 3) and intend to pursue this in future work. We think one promising approach could be based on off-policy methods [10], which applies to cases where the behaviour policy differs from the (evaluation) policy of interest (importance sampling is one simple example). In our case, the evaluation policy can still be random, whereas the behaviour policy is exploratory.
>
> Moreover, it is worth noting that the choice of an offline "base policy" is a recurring theme concerning model-based (and related) methods [9], and a random policy is a widespread choice [4-6, 11-13]. Some works use a mixture of a pre-trained and a random policy [7, 8], whereas some other works [5] attempt to expose and tackle the issue explicitly.
>
> (Continued in next post.)
>
> [1] Richemond & Maginnis 2017, "On Wasserstein Reinforcement Learning and the Fokker Planck equation." https://arxiv.org/abs/1712.07185
> [2] Sohl-Dickstein et al. 2015, "Deep Unsupervised Learning using Nonequilibrium Thermodynamics." https://arxiv.org/abs/1503.03585
> [3] Poole et al. 2016, "Exponential expressivity in deep neural networks through transient chaos." https://arxiv.org/abs/1606.05340
> [4] Savinov et al. 2018, "Semi-parametric Topological Memory for Navigation." https://arxiv.org/abs/1803.00653
> [5] Ha & Schmidhuber 2018, "World Models." https://arxiv.org/abs/1803.10122
> [6] Savinov et al. 2018, "Episodic Curiosity Through Reachability." https://arxiv.org/abs/1810.02274
> [7] Oh et al. 2015, "Action-Conditional Video Prediction using Deep Networks in Atari Games." https://arxiv.org/pdf/1507.08750.pdf
> [8] Chiappa et al. 2017, "Recurrent Environment Simulators." https://arxiv.org/abs/1704.02254
> [9] http://rail.eecs.berkeley.edu/deeprlcourse-fa17/f17docs/lecture_9_model_based_rl.pdf
> [10] Munos et al. 2016, "Safe and Efficient Off-Policy Reinforcement Learning." https://arxiv.org/abs/1606.02647
> [11] Nagabandi et al. 2017, "Learning Image-Conditioned Dynamics Models for Control of Under-actuated Legged Millirobots." https://arxiv.org/abs/1711.05253
> [12] Kulkarni et al. 2019, "Unsupervised Learning of Object Keypoints for Perception and Control." https://arxiv.org/abs/1906.11883
> [13] Anand et al. 2019, "Unsupervised State Representation Learning in Atari." https://arxiv.org/abs/1906.08226

---

> > ### Author Response · Authors · 2019-11-07
> > **First Response (Part 2/2)**
> >
> > (Continuation of the previous post.)
> >
> > > For example, can you use the learned arrow of time to solve the environments in (Krakovna et al), and how does it compare to relative reachability and its many variants?
> >
> > Thank you for the suggestion -- we will investigate this and get back to you.
> >
> > In the mean time, we remark that Krakovna et al. 2018 [11] propose to compare the reachability of all other states from a given state to that of a counterfactual "baseline state" that the system would be in had the agent been inactive. However, finding what this baseline state is requires counterfactual reasoning, which in turn assumes that a forward model of the environment dynamics is available. This is very unlike our method, which makes no such assumption (cf. p. 5 of our manuscript: "In contrast, Krakovna et al. (2018) propose [...]").
> >
> > > Similarly, how does your intrinsic reward compare to existing exploration methods (of which there are many, but consider count-based methods, curiosity (Pathak et al), random network distillation (Burda et al))
> >
> > By our assessment, the strengths and shortcomings of our intrinsic reward seem to differ from those of existing techniques. For instance, the class of curiosity methods that rely on the forward prediction error of a model are known to be susceptible to the so-called noisy-TV problem [12], where a local source of (unpredictable) stochasticity leads to a large prediction error and therefore a large reward. However, these methods are robust against temporally correlated sources of noise, because they can be predicted away by a powerful enough model. Meanwhile, our results in Fig 9 (App. C.1.1) show that the h-potential has the opposite tendency -- it is rather robust to uncorrelated sources of noise, but might get distracted by temporarily correlated noise. This hints towards potential synergies between our method and the various others in the literature, which we intend to explore in future work.
> >
> > > Consider for example the Overcooked environment [...]
> >
> > Thanks for the pointer, this is fascinating! We will respond to your suggestion in an upcoming post.
> >
> > [11] Krakovna et al. 2018, "Penalizing side effects using stepwise relative reachability." https://arxiv.org/abs/1806.01186
> > [12] Burda et al. 2018, "Exploration by Random Network Distillation." https://arxiv.org/abs/1810.12894

---

> > > ### Author Response · Authors · 2019-11-07
> > > **Overcooked (Part 3/2)**
> > >
> > > Once again, thanks for pointing us to the Overcooked environment!
> > >
> > > > One possibility is to redefine the arrow of time to be [...]
> > >
> > > Thank you for this rather interesting idea -- we summarise our high-level understanding of it. What you propose is to "bootstrap" a h-potential by (a) training it on a random policy, (b) improving the policy (or perhaps even an ensemble of policies) by training it to reduce the h-potential, (c) using the improved policy (or the ensemble) to gather more trajectories  and (d) refining the h-potential on the new trajectories, ad infinitum.
> > >
> > > Step (d) might, however, require an off-policy re-weighting via importance sampling or a more sophisticated method (as discussed in part 1). This is to prevent the h-potential from adapting to a particular agent (or a population of agents).
> > >
> > > We will try to investigate this further, albeit possibly in a simplified setting. Thanks again!

---

> > > > ### Author Response · Authors · 2019-11-08
> > > > **First Response to the Final Section of your Review (Part 4/2)**
> > > >
> > > > This is the final part of our (now 4 part) first response, and concerns the final section of your review ("This section did not affect my assessment of the paper").
> > > >
> > > > > Have you considered finding theoretically what it means to take the difference between the h-potential of two states (i.e. your reachability measure)?
> > > >
> > > > This is indeed a very fascinating avenue of research, one that we believe can benefit from concurrent research in statistical physics (cf. fluctuation theorems).
> > > >
> > > > > I could imagine that the answer is something like “reachability(s, s’) is proportional to the log probability of reaching s’ from s when acting according to a uniformly random policy”. Perhaps this has to be normalized against the log probability of reaching other states from s. This would be very interesting as a potential definition of reachability.
> > > >
> > > > Our intuitions seem aligned -- this is mere speculation at this point, but the quantity $h(s') - h(s)$ might be related to the log of the probability ratio of the forward ($s \to s'$) and the backward ($s' \to s$) trajectories, i.e. $\exp (h(s') - h(s)) \propto \frac{p(s \to s')}{p(s' \to s)}$, where the $\propto$ is in expectation. This is reminiscent of recent developments in non-equlibrium thermodynamics [13] (Eqn. 1 of [14] describes the matter succinctly). From the machine learning perspective, [15] is also quite interesting in this context.
> > > >
> > > > > Consider adding a discussion of ergodicity (as applied to Markov chains / MDPs), which will be more familiar to the audience and serves a similar purpose as the discussion on Hamiltonian systems.
> > > >
> > > > A short discussion on ergodicity can be found in p. 5 ("Observe that this condition much weaker than ergodicity [...]").
> > > >
> > > > The crux of the matter is that our method can be useful in both ergodic (e.g. Mountain Car, Pendulum) and non-ergodic environments (e.g. all 2D world environments we consider). In the latter, a notion of dissipation is somewhat apparent (it manifests itself e.g. when the plant dies in 2D world with drying tomatoes or a box is pushed against a wall in Sokoban), but the former warrants some extra caution. For instance, the Mountain Car environments we consider (with and without friction) are both ergodic, but one is not dissipative (without friction) whereas the other (with friction) is.
> > > >
> > > > The burden is hence on us to provide a prototypical example of systems where our method can be expected to fail, and one natural choice is a system where Liouville's theorem holds. Such systems have a known property that it is difficult to construct a Lyapunov functional that may function as an arrow of time (and can be learned by the h-potential), cf. [16, p. 279]. Despite the somewhat obscure reference (from the ML perspective), we think this is relevant discussion for the RL community, given how such systems often find use as benchmarking environments (Mountain-Car, (Double)Pendulums, etc).
> > > >
> > > > This concludes our first response. An update is underway; in the meantime, please don't hesitate to let us know if you think there is anything we can expand on.
> > > >
> > > > [13] Seifert 2012, "Stochastic thermodynamics, fluctuation theorems and molecular machines." https://iopscience.iop.org/article/10.1088/0034-4885/75/12/126001/pdf
> > > > [14] Gong & Quan 2015, "Jarzynski equality, Crooks fluctuation theorem, and the fluctuation theorems of heat for arbitrary initial states." https://journals.aps.org/pre/abstract/10.1103/PhysRevE.92.012131
> > > > [15] Goyal et al. 2017, "Variational Walkback: Learning a Transition Operator as a Stochastic Recurrent Net." https://arxiv.org/abs/1711.02282
> > > > [16] Prigogine 1977, "Nobel Lecture: Time, Structure and Fluctuations." https://www.nobelprize.org/prizes/chemistry/1977/prigogine/lecture/

---

> > > > > ### Comment · AnonReviewer3 · 2019-11-11
> > > > > **Thanks!**
> > > > >
> > > > > Thanks for the detailed response!
> > > > >
> > > > > > That said, we view our work as an attempt to view problems in RL from the lens of ideas and concepts that are already well understood in statistical physics.
> > > > >
> > > > > This makes sense, and I’m glad you are doing it :)
> > > > >
> > > > > A lot of the discussion in these comments have been particularly helpful for this. In particular, one distinction I failed to fully appreciate when reading the original paper is that it is “easier” for environments to be ergodic than to be Hamiltonian systems, since ergodic environments only require that _some_ action leads to every state, whereas in a Hamiltonian system _random_ actions need to make every state equally likely (excluding states that can never be visited).
> > > > >
> > > > > > In the mean time, we remark that Krakovna et al. 2018 [11] propose to compare the reachability of all other states from a given state to that of a counterfactual "baseline state" that the system would be in had the agent been inactive. However, finding what this baseline state is requires counterfactual reasoning, which in turn assumes that a forward model of the environment dynamics is available.
> > > > >
> > > > > This is a good point that I hadn’t considered. This probably means that you wouldn’t solve all of the environments there — probably your learned arrow of time doesn’t satisfy all of the desiderata that Krakovna et al propose. It would still be interesting to see which desiderata it does support, even without the ability to take counterfactuals.
> > > > >
> > > > > > This hints towards potential synergies between our method and the various others in the literature, which we intend to explore in future work.
> > > > >
> > > > > Very exciting, I look forward to it!
> > > > >
> > > > > > Thank you for this rather interesting idea -- we summarise our high-level understanding of it. What you propose is to "bootstrap" a h-potential by (a) training it on a random policy, (b) improving the policy (or perhaps even an ensemble of policies) by training it to reduce the h-potential, (c) using the improved policy (or the ensemble) to gather more trajectories  and (d) refining the h-potential on the new trajectories, ad infinitum.
> > > > >
> > > > > Yes, that’s right.
> > > > >
> > > > > > Step (d) might, however, require an off-policy re-weighting via importance sampling or a more sophisticated method (as discussed in part 1). This is to prevent the h-potential from adapting to a particular agent (or a population of agents).
> > > > >
> > > > > Yeah, probably something along these lines would be needed. Though I wouldn’t be surprised if you could just generate trajectories from a mixture of the current and older agents, and that was sufficient to get good results.
> > > > >
> > > > > > An update is underway; in the meantime, please don't hesitate to let us know if you think there is anything we can expand on.
> > > > >
> > > > > There’s nothing in particular; as I mentioned before, I think additional experiments (of several kinds) would be the best way to improve the paper.

---

> ### Author Response · Authors · 2019-11-13
> **Update Notification**
>
> Dear Reviewer 3, thanks again for your detailed review and speedy response. We’re happy to notify you that our manuscript has been updated to reflect your many suggestions that helped us improve our paper. To summarize the changes:
>
> - We improve the writing by pruning some of the jargon from the main text.
> - We include an appendix (C.1.4) showcasing results on the conveyor belt environment from Krakovna et al. 2018.
> - We include another appendix (C.3) where we experiment with your suggestion about bootstrapping the h-potential (albeit in a simple environment, due to the time and resource constraints in place).
>
> We hope to have changed your assessment of our work for the better; should that not be the case, please do not hesitate to get in touch with us!

---

> > ### Comment · AnonReviewer3 · 2019-11-13
> > **Interesting experiment**
> >
> > C.1.4 looks great to me :)
> >
> > C.3 is interesting -- I think the exploration bias shows a problem with using reachability as an intrinsic reward. Specifically, it incentivizes the agent to do the _most_ difficult / unreachable task, whereas usually with intrinsic rewards we would like the agent to do all of the difficult / unreachable tasks, even if they are varying levels of difficulty. (In the context of MountainCar, this would mean that the car climbs the left and right mountains, instead of just the right one.) It's not immediately apparent to me how to fix the issue, but probably it's doable.
> >
> > I don't completely understand your random policy solution, but it looks like it throws away the information in the h-potential, which is a shame. In more complex environments, there will be lots of "uninteresting" directions, "somewhat difficult" directions, and "most difficult" directions. Optimizing for the h-potential will get only the most difficult direction, but if you throw away the information in the h-potential, then you explore in all three types of directions -- ideally you would only explore in the latter two directions.
> >
> > (Obviously this is all a suggestion for future work, since there isn’t enough time to update the paper now. I mention it because I’m excited for future work along these lines.)
> >
> > I'd recommend adding in the point about how side effects depend on counterfactuals to Section 3.2, and the issue with using the h-potential as an intrinsic reward to Section 3.3 (while leaving the experiments in the appendix) since they both add conceptual clarity.

---

> > > ### Author Response · Authors · 2019-11-14
> > > **Thanks! We have updated our manuscript again.**
> > >
> > > We are glad you like C.1.4. :)
> > >
> > > > I think the exploration bias shows a problem with using reachability as an intrinsic reward [...]
> > >
> > > This is an interesting way to look at curiosity in general — the dichotomy between diversity and the drive to reach difficult-to-reach states appears to be underexplored. In this context, the strength of our method seems to be that it can "convert" diversity (e.g. due to uniform initialization over state space in the extreme case) to a measure of (un-)reachability. But ideally, a good exploration strategy would need both.
> > >
> > > > I don’t completely understand your random policy solution, but it looks like it throws away the information in the h-potential, which is a shame.
> > >
> > > In a nutshell, we pretend that the h-potential is a random neural network, and train the exploratory policy to minimize it. This is repeated a few times over, and at every iteration, the fake h-potential attracts the exploratory policy to different (diverse) regions in the state space. All such policies taken together end up providing some of the needed diversity. This should be vaguely reminiscent of Random Network Distillation proposed in Burda et al.[12].
> > >
> > > All that said, we agree it would be better if we could actually use the learned h-potential for this (it indeed seems possible to craft an algorithm that uses the reachability signal from the h-potential and combines it with something that promotes diversity).
> > >
> > > > I’d recommend adding in the point about how side effects depend on counterfactuals to Section 3.2
> > >
> > > The newest revision expands on this. We remark that the concept of side-effects as defined in Leike et al. [17] does not seem to directly depend on counterfactuals (instead, it relies on reversibility — cf. page 5 of [17]). Rather, it is the method proposed in Krakovna et al. [11] that does. Eysenbach et al. [18], for instance, describe yet another method that takes a somewhat orthogonal (but interesting) approach.
> > >
> > > > the issue with using the h-potential as an intrinsic reward to Section 3.3 (while leaving the experiments in the appendix) since they both add conceptual clarity.
> > >
> > > Of course — we mention this in the latest revision.
> > >
> > > Thanks once again!
> > >
> > > [17] Leike et al. 2017, “AI-Safety Gridworlds.” https://arxiv.org/abs/1711.09883
> > > [18] Eysenbach et al. 2017, “Leave no Trace: Learning to Reset for Safe and Autonomous Reinforcement Learning.” https://arxiv.org/abs/1711.06782

---

### Author Response · Authors · 2019-11-13
**Change Log (First Revision)**

We have updated our manuscript to reflect the many helpful suggestions made by the reviewers. In summary,

- We include an appendix (C.1.4) to showcase additional results on a 2D world environment (Reviewer 3).
- We include an appendix (C.3) where we experiment with one of reviewer 3's suggestions.
- We have improved the writing and clarified minor details.

---

### Decision · Program_Chairs · 2019-12-19

**Decision:**

Accept (Poster)

**Comment:**

This paper develops the notion of the arrow of time in MDPs and explores how this might be useful in RL. All the reviewers found the paper thought provoking, well-written, and they believe the work could have significant impact. The paper does not fit the typical mold: it presents some ideas and uses illustrative experiments to suggest the potential utility of the arrow without nailing down a final algorithm or make a precise performance claim. Overall it is a solid paper, and the reviewers all agreed on acceptance.

There are certainly weaknesses in the work, and there is a bit of work to do to get this paper ready. R2 had a nice suggestion of a baseline based on simply learning a transition model (its described in the updated review)---please include it. The description of the experimental methodology is a bit of a mess. Most of the experiments in the paper do not clearly indicate how many runs were conducted or how errorbars where computed or what they represent.  It is likely that only a handful of runs were used, which is surprising given the size of some of the domains used. In many cases the figure caption does not even indicate which domain the data came from. All of this is dangerously close to criteria for rejection; please do better.

Readability is also known as empowerment and it would be good to discuss this connection. In general the paper was a bit light on connections outlining how information theory has been used in RL. I suggest you start here (http://www2.hawaii.edu/~sstill/StillPrecup2011.pdf) to improve this aspect. Finally, the paper has a very large appendix (~14 oages) with many many more experiments and theory. I am still not convinced that the balance is quite right. This is probably a journal or long arxiv paper. Maybe this paper should be thought of as a nectar version of a longer standalone arxiv paper.

Finally, relying on effectiveness of random exploration is no small thing and there is a long history in RL of ideas that would work well, given it is easy to gather data that accurately summarizes the dynamics of the world (e.g. proto-value, funcs). Many ideas are effective given this assumption. The paper should clearly and honestly discuss this assumption, and provide some arguments why there is hope.